# HelmSim: Learning Helmholtz Dynamics for Interpretable Fluid Simulation

## Abstract

Fluid simulation is a long-standing challenge due to the intrinsic high-dimensional non-linear dynamics. Previous methods usually utilize the non-linear modeling capability of deep models to directly estimate velocity fields for future prediction. However, skipping over inherent physical properties but directly learning superficial velocity fields will overwhelm the model from generating precise or physics-reliable results. In this paper, we propose the *HelmSim* toward an accurate and interpretable simulator for fluid. Inspired by the Helmholtz theorem, we design a *HelmDynamic* block to learn the *Helmholtz dynamics*, which decomposes fluid dynamics into more solvable curl-free and divergence-free parts, physically corresponding to potential and stream functions of fluid. By embedding the HelmDynamic block into a *Multiscale Integration Network*, HelmSim can integrate learned Helmholtz dynamics along temporal dimension in multiple spatial scales to yield future fluid. Comparing with previous velocity estimating methods, HelmSim is faithfully derived from Helmholtz theorem and ravels out complex fluid dynamics with physically interpretable evidence. Experimentally, our proposed HelmSim achieves the consistent state-of-the-art in both numerical simulated and real-world observed benchmarks, even for scenarios with complex boundaries.

## 1 Introduction

Fluid is one of basic substances in the physical world. Its simulation is with immense importance in extensive real-world applications, such as atmospheric simulation for weather forecasting and airflow modeling for airfoil design, which has attracted significant attention from both science and engineering areas. However, it is quite challenging to capture the intricate high-dimensional non-linear dynamics within fluid due to the imperfect observations, coupled multiscale interactions, etc.

Recently, deep models have achieved impressive progress in solving complex physical systems (Karniadakis et al., 2021; Wang et al., 2023). One paradigm is learning neural operators to directly predict the future fluid field based on past observations (Lu et al., 2021a; Li et al., 2021; Wu et al., 2023). These methods focus on leveraging the non-linear modeling capacity of deep models to approximate complex mappings between past and future fluids. However, directly learning neural operators may fail in generating interpretable evidence for prediction results and incurring uncontrolled errors. Another mainstreaming paradigm attempts to estimate the dynamic fields of fluid with deep models for future simulation. It is notable that the superficial dynamics are actually driven by underlying physical rules. Directly estimating the velocity fields regarding less physical properties may overwhelm the model from generating precise and plausible simulation results (Sun et al., 2018; Zhang et al., 2022). As shown in Figure 1, it is hard to directly capture the complex dynamics of fluid, where the learned dynamics will be too tanglesome to guide the fluid simulation.

To tackle the above challenges, we attempt to capture the intricate dynamics with physical insights for accurate and interpretable fluid simulation. In this paper, we dive into the physical properties of fluid and propose the *Helmholtz dynamics* as a new paradigm to represent fluid dynamics. Concretely, Helmholtz dynamics is inspired from the Helmholtz theorem (Bladel, 1959) and attributes the intricate dynamics into the potential and stream functions of fluid, which are intrinsic physical quantities of fluid and can directly derive the curl-free and divergence-free parts of fluid respectively. Comparing with superficial velocity fields, our proposed Helmholtz dynamics decompose the intricate dynamics into more solvable components, thereby easing the dynamics learning pro-

Figure 1: Comparison on dynamics learning and fluid prediction. Different from the numerical method (Ruzanski et al., 2011) and optical-flow-based deep model (Sun et al., 2018), our proposed HelmSim infers the dynamics from the inherent physics quantities: potential and stream functions.

cess of deep models. Besides, this new dynamics requires the model to learn inherent properties of fluid explicitly, which also empowers the simulation with endogenetic physical interpretability.

Based on the above insights, we present the *HelmSim* model with *HelmDynamic* blocks to capture the Helmholtz dynamics for interpretable fluid simulation. HelmDynamic is faithfully implemented from the Helmholtz decomposition, which can separately estimate the potential and stream functions of fluid from learned spatiotemporal correlations and further derive curl-free and divergence-free velocities. As a flexible module, HelmDynamic can conveniently encode boundary conditions into the correlation calculation process and adapt to complex boundary settings in multifarious real-world applications. Further, we design the *Multiscale Integration Network* in HelmSim to fit the multiscale nature of fluid, which can integrate Helmholtz dynamics learned by HelmDynamic blocks along temporal dimension in multiple spatial scales to predict the future fluid. Experimentally, HelmSim achieves the consistent state-of-the-art in various scenarios, covering both synthetic and real-world benchmarks with complex boundary settings. Our contributions are summarized in the following:

- Inspired by the Helmholtz theorem, we propose the *Helmholtz dynamics* to attribute intricate dynamics into inherent properties of fluid, which decomposes intricate dynamics into more solvable parts and empowers the simulation process with physical interpretability.

- We propose the *HelmSim* with the *HelmDynamic block* to capture Helmholtz dynamics. By integrating learned dynamics along temporal dimension through *Multiscale Integration Network*, HelmSim can predict the future fluid with physically interpretable evidence.

- HelmSim achieves consistent state-of-the-art in extensive benchmarks, covering both synthetic and real-world datasets, as well as various boundary conditions.

## 2 PRELIMINARIES

### 2.1 FLUID SIMULATION

As a foundation problem in science and engineering areas, fluid simulation has been widely explored. Traditional methods can solve Navier-Stokes equations with numerical algorithms, while they may fail in the real-world fluid due to imperfect observations of initial conditions and inaccurate estimation of equation parameters. Besides, these numerical methods also suffer from the huge computation cost. Recently, owing to the great non-linear modeling capacity, data-driven deep models for fluid simulation have attached substantial interests, which can be roughly categorized into the following paradigms according to whether learning velocity fields explicitly or not.

**Neural fluid simulator** This paradigm of works attempts to directly generate future fluid with deep models. One direction is formalizing partial differential equations (PDEs), initial and boundary conditions as loss function terms, and parameterizing the solution as a deep model (Evans, 2010; Raissi et al., 2019; 2020; Lu et al., 2021b). These approaches rely highly on exact physics equations, thereby still suffering from imperfect observations and inherent randomness in real-world applications. Another branch of methods does not require the exact formalization of governing PDEs. They attempt to learn neural operators to approximate complex input-output mappings in scientific tasks, which enables the prediction of future fluid solely based on past observations. For example, Lu et al. (2021a) proposed the DeepONet in a branch-trunk framework with proved universal approximation capability. FNO (Li et al., 2021) approximates the integral operator through a linear transformation within the Fourier domain. Afterward, U-NO (Rahman et al., 2023) enhances FNO with the multiscale framework. Later, Wu et al. (2023) proposed latent spectral models to solve high-dimensional

PDEs in the latent space by learning multiple basis operators. However, these methods may fail to provide interpretable evidence for prediction results, such as intuitive physics quantities or visible velocity fields. Going beyond the above-mentioned methods, we propose HelmSim as a purely data-driven model but with special designs to enhance physical interpretability.

**Fluid dynamics modeling**   Estimating velocity fields is a direct and intuitive way for predicting the future fluid. Typically, optical flow (Horn & Schunck, 1981) is proposed to describe the motion between two successive observations. Recently, many deep models have been proposed to estimate optical flow, such as PWC-Net (Sun et al., 2018) and RAFT (Teed & Deng, 2020). However, since the optical flow was originally designed for rigid bodies, it struggles seriously in capturing fluid motion and will bring serious accumulation errors in the prediction process. Especially for fluid, Zhang et al. (2022) incorporate physical constraints from Navier-Stokes equations to refine the velocity field predicted by PWC-Net (Sun et al., 2018) and further embed the advection-diffusion equation into the deep model to predict the future fluid. Recently, Deng et al. (2023) ensemble the observable Eulerian flow and the hidden Lagrangian vortical evolution to capture the intricate dynamics in the fluid. Unlike previous, we propose to learn the inherent physics quantities of fluid for Helmholtz dynamics and further predict the future fluid with temporal integration, which decomposes the intricate dynamics into more solvable components and facilitates our model with physical interpretability.

**Computer graphic for fluid simulation**   Solving Navier-Stokes equations in computer graphics with machine learning techniques often employs a stream function paradigm to enforce the incompressibility condition (Ando et al., 2015). Kim et al. (2019) successfully synthesizes plausible and divergence-free 2D and 3D fluid velocities from a set of reduced parameters and ground truth velocity supervision, a rarity in real-world data. Recently, Liu et al. (2021) estimate the underlying physics of advection-diffusion equations, incorporating ground truth velocity and diffusion tensors supervision. Franz et al. (2023) simulate a realistic 3D density and velocity sequence from single-view sequences without 3D supervision, but it is not designed for predictive tasks as it utilizes future information to calculate current density. Unlike previous methods, our proposed method learns the velocity solely from observed physics through Helmholtz dynamics, rather than single stream function, in an end-to-end fashion without ground truth velocity supervision, which enables our model to capture more intricate fluid dynamics and extends its applicability to a broader range of scenarios.

## 2.2   HELMHOLTZ DECOMPOSITION

In fluid dynamics, the Helmholtz decomposition (Bladel, 1959) plays an important role, which can decompose a dynamic field into a curl-free component and a divergence-free component for simplification, and is highly related to the solvability theory of Navier-Stokes equations (Faith A., 2013).

Given a 3D dynamic field $\mathbf{F} : \mathbb{V} \rightarrow \mathbb{R}^3$ with a bounded domain $\mathbb{V} \subseteq \mathbb{R}^3$, we can obtain the following decomposition with the Helmholtz theorem:

$$\mathbf{F}(\mathbf{r}) = \nabla\Phi(\mathbf{r}) + \nabla \times \mathbf{A}(\mathbf{r}), \mathbf{r} \in \mathbb{V}. \tag{1}$$

It is notable that $\Phi : \mathbb{V} \rightarrow \mathbb{R}$ denotes the *potential function*, which is a scalar field with its gradient field $\nabla\Phi$ representing the curl-free part of $\mathbf{F}$ guaranteed by $\nabla \times (\nabla\Phi) = \mathbf{0}$. And $\mathbf{A} : \mathbb{V} \rightarrow \mathbb{R}^3$ is named as the *stream function*, which is a vector field with $\nabla \times \mathbf{A}$ represents the divergence-free part of $\mathbf{F}$ underpinned by $\nabla(\nabla \times \mathbf{A}) = \mathbf{0}$, thereby also indicating the incompressibility of the flow field.

Following previous well-acknowledged works and conventional settings in this area (Li et al., 2021), we focus on the 2D fluid simulation in this paper and project the Helmholtz theorem into the 2D space by restricting the $z$-axis component of $\mathbf{F}$ to be 0, i.e. $\mathbf{F}(\mathbf{r}) = (\mathbf{F}_x(\mathbf{r}), \mathbf{F}_y(\mathbf{r}), 0)^\mathsf{T}$. This reduction causes the vanishing of components of the stream function along the $x$-axis and $y$-axis, namely $\mathbf{A}(\mathbf{r}) = ((0, 0, \mathbf{A}_z(\mathbf{r}))^\mathsf{T}$, indicating that the stream function is degenerated to a scalar field.

Inspired by the above theorem, we first define the Helmholtz dynamics to represent fluid dynamics and faithfully implement Helmholtz decomposition as a built-in block, i.e. HelmDynamic block.

## 3   HELMSIM

As aforementioned, we highlight the key components of fluid simulation as providing physical interpretability and handling intricate dynamics. To achieve these objectives, we present the HelmSim

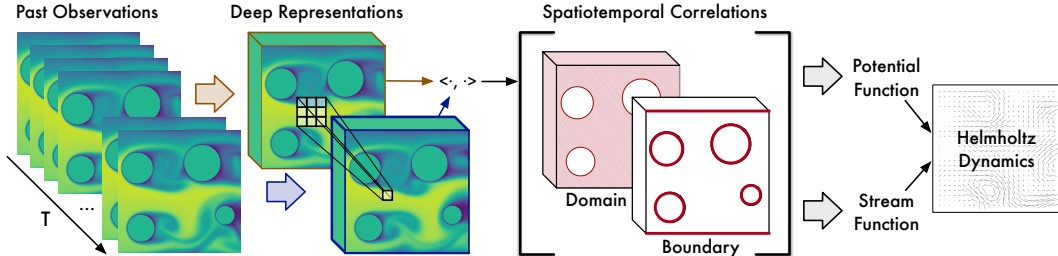

Figure 2: HelmDynamic block, which learns spatiotemporal correlations to estimate potential and stream functions of fluid from past observations for simulating the Helmholtz dynamics.

model with *HelmDynamic* blocks to capture the *Helmholtz dynamics* for 2D fluid, which is inspired from the Helmholtz theorem and attributes superficial complex dynamics into the inherent properties of fluid. Further, we design the *Multiscale Integration Network* to integrate the learned dynamics along the temporal dimension in multiple scales to attain the future fluid.

### 3.1 LEARNING HELMHOLTZ DYNAMICS

To tackle intricate fluid dynamics, we propose the HelmDynamic block to learn Helmholtz dynamics instead of directly learning velocity fluid, which is faithfully implemented from the Helmholtz theorem and can decompose the complex dynamics into more solvable components.

**Helmholtz dynamics for 2D fluid**  According to the Helmholtz theorem (Eq. 1), the fluid dynamics can be equivalently decomposed into *curl-free* and *divergence-free* parts for simplification. Thus, we define Helmholtz dynamics as the function of potential and stream functions, which are inherent physics quantities of fluid. Concretely, for a 2D fluid defined in the domain $\mathbb{V} \subseteq \mathbb{R}^2$, its Helmholtz dynamics $\mathbf{F}_{\text{Helm}}$ can be formalized by potential function $\Phi : \mathbb{V} \to \mathbb{R}$ and stream function $\mathbf{A} : \mathbb{V} \to \mathbb{R}$ of fluid as follows:

$$\mathbf{F}_{\text{Helm}}(\Phi, \mathbf{A}) = \nabla\Phi + \nabla \times \mathbf{A} = \underbrace{\left(\frac{\partial\Phi}{\partial x}, \frac{\partial\Phi}{\partial y}\right)}_{\text{Curl-free Velocity}} + \underbrace{\left(\frac{\partial\mathbf{A}}{\partial y}, -\frac{\partial\mathbf{A}}{\partial x}\right)}_{\text{Divergence-free Velocity}} . \tag{2}$$

Note that according to the Helmholtz theorem (Eq. 1), the function value of $\mathbf{F}_{\text{Helm}}$ is equivalent to the real dynamic field $\mathbf{F}$ but is more tractable. By incorporating $\Phi$ and $\mathbf{A}$, Helmholtz dynamics naturally decomposes the intricate fluid into more solvable components and ravels out the complex dynamics into intrinsic physics quantities, thus benefiting the dynamics modeling (Bhatia et al., 2013).

**HelmDynamics block**  To obtain the Helmholtz dynamics, we propose the HelmDynamic block to estimate the potential and stream functions from past observations. Technically, as shown in Figure 2, we first embed input observations into two successive deep representations to keep the temporal dynamics information explicitly. Given a sequence of successively observed 2D fluid $\mathbf{x} = [\mathbf{x}_1, \cdots, \mathbf{x}_T], \mathbf{x}_i \in \mathbb{R}^{H\times W}$, this process can be formalized as:

$$\widehat{\mathbf{x}}_{T-1} = \text{Embed}\left(\mathbf{x}_{1:(T-1)}\right)$$
$$\widehat{\mathbf{x}}_T = \text{Embed}\left(\mathbf{x}_{2:T}\right), \tag{3}$$

Figure 3: Transform potential and stream functions into the velocity field.

where $\widehat{\mathbf{x}}_{T-1}, \widehat{\mathbf{x}}_T \in \mathbb{R}^{d_{\text{model}}\times H\times W}$ and the temporal dimension of $(T-1)$ observations are projected to the channel dimension $d_{\text{model}}$ by two convolutional layers with an in-between activation function.

Next, following the convention in dynamics modeling (Sun et al., 2018; Teed & Deng, 2020), we adopt spatiotemporal correlations between fluid at the previous timestamp and the current timestamp to represent the dynamics information. Especially as physics quantities of fluid are highly affected

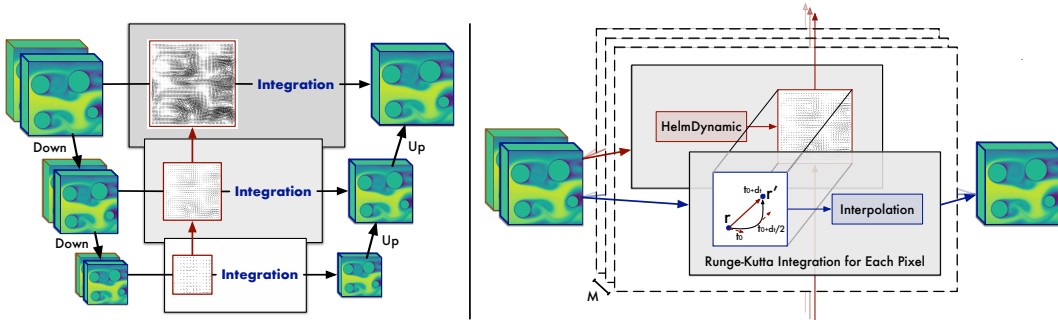

Figure 4: Multiscale Integration Network (left part), which integrates learned Helmholtz dynamics of the fluid along the temporal dimension (right part) in multiple scales to generate future fluid field.

by boundary conditions, we also include boundary conditions $\mathbb{S}$ into the calculation of correlations:

$$\mathbf{c}(\mathbf{r}) = \text{Concat}\left(\left[\langle \widehat{\mathbf{x}}_T(\mathbf{r}), \widehat{\mathbf{x}}_{T-1}(\mathbf{r}')\rangle\right]_{\mathbf{r}'\in\mathbb{V}_\mathbf{r}}, \left[\mathbb{1}_\mathbb{S}(\mathbf{r}')\langle \widehat{\mathbf{x}}_T(\mathbf{r}), \widehat{\mathbf{x}}_{T-1}(\mathbf{r}')\rangle\right]_{\mathbf{r}'\in\mathbb{V}_\mathbf{r}}\right), \ \mathbf{r}\in\mathbb{V} \quad (4)$$

where $\langle\cdot,\cdot\rangle$ denotes the inner-product operation and $\mathbb{V}_\mathbf{r}$ denotes the neighbors around position $\mathbf{r}$. $\mathbb{1}_\mathbb{S}(\cdot)$ denote the indicator function, whose value is 1 when $\mathbf{r}'\in\mathbb{S}$ and 0 otherwise. $\mathbf{c}(\mathbf{r})\in\mathbb{R}^{2|\mathbb{V}_\mathbf{r}|}$ represents the correlation map between the current fluid at $\mathbf{r}$ and its $|\mathbb{V}_\mathbf{r}|$ neighbors in the previous fluid, with additional consideration on the boundary conditions $\mathbb{S}$. Thus, we obtain the extracted dynamics information $\mathbf{c}\in\mathbb{R}^{2|\mathbb{V}_\mathbf{r}|\times H\times W}$. Subsequently, we can decode the potential and stream functions from the dynamics information and calculate the Helmholtz dynamics as follows:

$$\widehat{\Phi} = \text{Decoder}_\Phi\left(\mathbf{c}\right), \ \widehat{\mathbf{A}} = \text{Decoder}_\mathbf{A}\left(\mathbf{c}\right), \ \widehat{\mathbf{F}}_{\text{Helm}} = \nabla\widehat{\Phi} + \nabla\times\widehat{\mathbf{A}}, \quad (5)$$

where $\widehat{\Phi}, \widehat{\mathbf{A}}\in\mathbb{R}^{H\times W}$ and $\widehat{\mathbf{F}}_{\text{Helm}}\in\mathbb{R}^{2\times H\times W}$ represent the learned 2D fields of curl-free velocity, divergence-free velocity, and combined velocity respectively (Figure 3). $\text{Decoder}_\Phi$ and $\text{Decoder}_\mathbf{A}$ are learnable deep layers instantiated as two convolutional layers with an in-between activation function. We summarize the above process as $\widehat{\mathbf{F}}_{\text{Helm}} = \text{HelmDynamic}(\widehat{\mathbf{x}}_{(T-1)}, \widehat{\mathbf{x}}_T)$ for conciseness.

## 3.2 MULTISCALE INTEGRATION NETWORK

After tackling intricate dynamics with the HelmDynamic block, we further present the Multiscale Integration Network to fuse the learned dynamics along the temporal dimension for predicting the future fluid, which consists of a multihead integration block and a multiscale modeling framework.

**Multihead integration** To simulate the complex dynamics in fluid, we employ a multihead design for temporal integration, which is widely used in the attention mechanism to augment the non-linear capacity of deep models (Vaswani et al., 2017). As shown in Figure 4, given two successive deep representations $\widehat{\mathbf{x}}_{(T-1)}, \widehat{\mathbf{x}}_T\in\mathbb{R}^{d_{\text{model}}\times H\times W}$ of fluid, we can firstly split them along the channel dimension for multiple heads and obtain $\widehat{\mathbf{x}}_{(T-1),i}, \widehat{\mathbf{x}}_{T,i}\in\mathbb{R}^{\frac{d_{\text{model}}}{M}\times H\times W}, i\in\{1,\cdots,M\}$, where $M$ is a hyperparameter. Then we compute multiple Helmholtz dynamics from multihead representations:

$$\left[\widehat{\mathbf{F}}_{\text{Helm},i}\right]_{i=1,\cdots,M} = \left[\text{HelmDynamic}(\widehat{\mathbf{x}}_{(T-1),i}, \widehat{\mathbf{x}}_{T,i})\right]_{i=1,\cdots,M}, \quad (6)$$

where $\widehat{\mathbf{F}}_{\text{Helm},i}\in\mathbb{R}^{2\times H\times W}$. For conciseness, we omit the head index $i$ and summarize the above process by $\widehat{\mathbf{F}}_{\text{Helm}} = \text{Multi-HelmDynamic}(\widehat{\mathbf{x}}_{(T-1)}, \widehat{\mathbf{x}}_T)$, where $\widehat{\mathbf{F}}_{\text{Helm}}\in\mathbb{R}^{M\times 2\times H\times W}$.

With these learned Helmholtz dynamics fields, we can estimate the future position of each pixel with numerical integration methods. Especially, for a position $\mathbf{r}$, we take the second-order Runge-Kutta method (DeVries & Wolf, 1994) to estimate its position in the future $\text{d}t$ time as shown in Figure 4, which can be formalized as $\mathbf{r}'_i = \mathbf{r} + \widehat{\mathbf{F}}_{\text{Helm},i}(\mathbf{r} + \widehat{\mathbf{F}}_{\text{Helm},i}(\mathbf{r})\frac{\text{d}t}{2})\text{d}t$. The above equation can directly deduce the next step representation by moving the pixel at $\mathbf{r}$ to $\mathbf{r}'$. Here we adopt the back-and-forth error compensation and correction (BFECC, 2005) method to move representations and interpolate them into a regular grid. We summarize the above numerical process as:

$$\widehat{\mathbf{x}}_{(T+1)} = \text{Integration}\left(\widehat{\mathbf{x}}_T, \widehat{\mathbf{F}}_{\text{Helm}}\right) = \text{Concat}\left(\left[\text{BFECC}\left(\widehat{\mathbf{x}}_{T,i}, \widehat{\mathbf{F}}_{\text{Helm},i}\right)\right]_{i=1,\cdots,M}\right). \quad (7)$$

**Multiscale modeling** It is known in physics that the fluid exhibits different properties at different scales. These multiscale dynamics mix and entangle with each other, making the fluid dynamics extremely intractable. Thus, we adopt the multiscale modeling framework in HelmSim.

For fluid embeddings $\widehat{\mathbf{x}}_{(T-1)}, \widehat{\mathbf{x}}_T \in \mathbb{R}^{d_{\text{model}} \times H \times W}$, we adopt the multiscale encoder to obtain deep representations in $L$ scales $\widehat{\mathbf{x}}_{(T-1)}^l, \widehat{\mathbf{x}}_T^l \in \mathbb{R}^{d_{\text{model}}^l \times \lfloor \frac{H}{2^{(l-1)}} \rfloor \times \lfloor \frac{W}{2^{(l-1)}} \rfloor}, l \in \{1, \cdots, L\}$. As the dynamics from larger scales is less affected by noise and can provide a reliable background velocity field for the small scales, we ensemble the learned dynamics from coarse to fine to ease the multiscale dynamics modeling process as shown in Figure 4. Overall, the prediction process in the $l$-th scale is

$$\widehat{\mathbf{F}}_{\text{Helm}}^l = \text{Multi-HelmDynamic} \left( \widehat{\mathbf{x}}_{(T-1)}^l, \widehat{\mathbf{x}}_T^l \right) + 2 \times \widehat{\mathbf{F}}_{\text{Helm}}^{(l+1)}$$
$$\widehat{\mathbf{x}}_{(T+1)}^l = \text{Integration} \left( \widehat{\mathbf{x}}_T^l, \widehat{\mathbf{F}}_{\text{Helm}}^l \right), \tag{8}$$

where $l$ is processed from $L$ to 1 and $\widehat{\mathbf{F}}_{\text{Helm}}^{(L+1)} = \mathbf{0}$. In each up-scaling, a scale coefficient of 2 is multiplied to the coarser-scale dynamics to align the velocity values. Eventually, we progressively aggregate the learned Helmholtz dynamics from large to small scales and obtain the final prediction of the fluid field with a projection layer. More details are deferred to Appendix A.2.

## 4 EXPERIMENTS

We extensively evaluate our proposed HelmSim on four benchmarks, including both simulated and real-world observed scenarios, covering known and unknown boundary settings (see Figure 5).

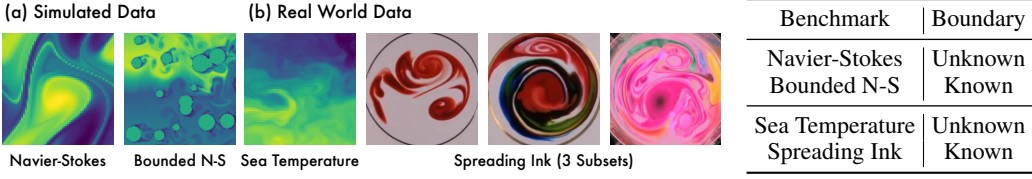

| Benchmark | Boundary |
|---|---|
| Navier-Stokes Bounded N-S | Unknown Known |
| Sea Temperature Spreading Ink | Unknown Known |

Figure 5: Summary of four experiment benchmarks, including (a) simulated and (b) real-world data.

**Baselines** We compare our proposed HelmSim with eight competitive baselines, including one numerical method DARTS (Ruzanski et al., 2011), four neural fluid simulators: LSM (Wu et al., 2023), U-NO (Rahman et al., 2023), WMT (Gupta et al., 2021), FNO (Li et al., 2021), two fluid-dynamics-modeling solutions: Vortex (Deng et al., 2023), PWC-Net with fluid Refinement (Zhang et al., 2022) and one vision backbone widely-used in AI4Science: U-Net (Ronneberger et al., 2015). Here, LSM and U-NO are previous state-of-the-art models in fluid simulation. Note that due to the inconsistent settings in fluid simulation, some of the baselines are not suitable for all benchmarks. Thus, in the main text, we only provide comparisons to the baselines on their official benchmarks. But to ensure transparency, we also provide the comprehensive results for all baselines in Table 15.

**Implementations** For fairness, we adopt the relative L2 as the loss function and train all the models for the same epochs in each benchmark, specifically, 500 epochs for Navier-Stokes (N-S), 200 epochs for Bounded N-S and Spreading Ink, 50000 iterations for Sea Temperature. For all benchmarks, we report the relative L2 in the test set for comparison. In addition, we also use the VGG perceptual loss (Johnson et al., 2016) for Spreading Ink to compare the visual quality of predictions. A comprehensive description is provided in Appendix A.

### 4.1 SIMULATED DATA

**Navier-Stokes with unknown boundary** This dataset is simulated from a viscous, incompressible fluid field on a two-dimensional unit torus, which obeys Navier-Stokes equations (Li et al., 2021). The task is to predict the future 10 steps based on the past 10 observations. To verify the model capacity in different resolutions, we generate three subsets ranging from $64 \times 64$ to $256 \times 256$ with 1000 training sequences, 200 validation sequences and 200 test sequences.

As presented in table 1, HelmSim significantly surpasses other models, demonstrating its advancement in fluid simulation. Especially in comparison with the second best model, HelmSim achieves

Figure 6: Showcase study on the Navier-Stokes dataset under the $64 \times 64$ input resolution. We also visualize the Helmholtz dynamics learned by HelmSim.

12.1% relative error reduction (0.1261 vs. 0.1435) in the $64 \times 64$ resolution setting and achieves consistent state-of-the-art in all time steps. Besides, HelmSim performs best for the inputs under various resolutions, verifying its capability to handle the dynamics in different scales.

To intuitively present the model capability, we also provide several showcases in Figure 6. In comparing to U-NO and LSM, we can find that HelmSim precisely predicts the fluid motion, especially the twist parts, which involve complex interactions among several groups of fluid particles. Besides, HelmSim can also generate the learned velocity field for each step, which reflects the rotation and diffusion of fluid, empowering simulation with interpretable evidence. These results demonstrate the advantages of HelmSim in capturing complex dynamics and endowing model interpretability.

**Bounded N-S with known boundary**   In real-world applications, we usually need to handle the complex boundary conditions in fluid simulation. To verify the model capacity in this setting, we adopt Taichi (Hu et al., 2019) engine to generate 2D fluid sequences with complex boundaries. Specifically, the generated dataset simulates a wide pipe scenario, where the incompressible fluid moves from left to right, passing by several solid columns. Each sequence is started at a random initial condition with resolution of $128 \times 128$. We set an appropriate viscosity and generate columns of different sizes. Note that this task contains the Karmen vortex phenomenon (Bayındır

Table 2: Model performance comparison on the Bounded N-S dataset.

| Model | Relative L2 |
|---|---|
| DARTS (Ruzanski et al., 2011) | 0.1820 |
| U-Net (Ronneberger et al., 2015) | 0.0846 |
| FNO (Li et al., 2021) | 0.1176 |
| MWT (Gupta et al., 2021) | 0.1407 |
| U-NO (Rahman et al., 2023) | 0.1200 |
| LSM (Wu et al., 2023) | 0.0737 |
| HelmSim (Ours) | **0.0652** |

& Namlı, 2021) with many vortices of various sizes, making this problem extremely challenging. We need to predict the future 10 steps based on the past 10 observations.

HelmSim also performs best in this challenging task and presents a consistent advantage in all prediction steps. Although U-Net is kind of close to HelmSim in averaged relative L2, it fails to capture the Karmen vortex phenomenon and results in blurry predictions (Figure 7), which will seriously impede its practicability. In contrast, HelmSim precisely predicts the Karmen vortex around boundaries with eidetic texture. This result is benefited from the Helmholtz dynamics modeling paradigm.

Besides, we also provide the comparison of learned dynamics among HelmSim, DARTS (2011) and PWC-Net (2022) in Figure 1. It is impressive that HelmSim can still capture the dynamics accurately, even for vortices around solid columns. This capability may come from that instead of directly learning velocities, HelmSim learns the potential and stream functions, thereby raveling

Table 1: Performance comparison on the Navier-Stokes dataset under different resolutions. Relative L2 is recorded. For clarity, the best result is in bold and the second best is underlined. The timewise error curve is recorded from the $64 \times 64$ settings.

| Model | $64 \times 64$ | $128 \times 128$ | $256 \times 256$ |
|---|---|---|---|
| DARTS (Ruzanski et al., 2011) | 0.8046 | 0.7002 | 0.7904 |
| U-Net (Ronneberger et al., 2015) | 0.1982 | 0.1589 | 0.2953 |
| FNO (Li et al., 2021) | 0.1556 | 0.1028 | 0.1645 |
| MWT (Gupta et al., 2021) | 0.1586 | 0.0841 | 0.1390 |
| U-NO (Rahman et al., 2023) | 0.1435 | 0.0913 | 0.1392 |
| LSM (Wu et al., 2023) | 0.1535 | 0.0961 | 0.1973 |
| HelmSim (Ours) | **0.1261** | **0.0807** | **0.1310** |

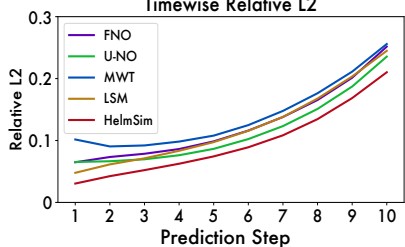

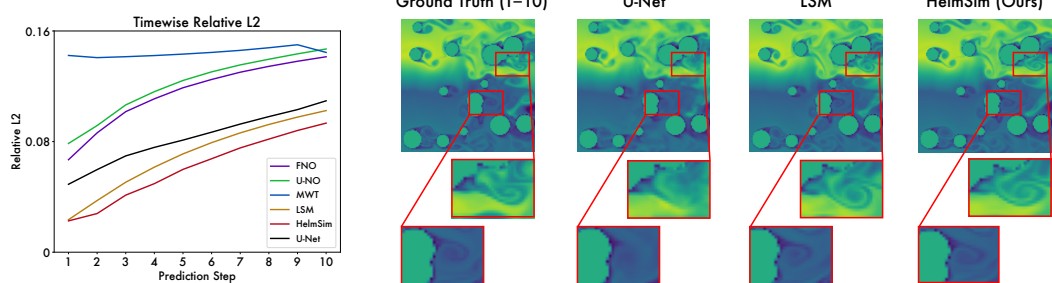

Figure 7: Timewise error and showcases on the Bounded N-S dataset. For clarity, we highlight and zoom in the key parts of fluid in red boxes.

out the intricate dynamics naturally. It is also notable that numerical method DARTS degenerates seriously in both quantitative results (Table 2) and learned dynamics (Figure 1), which highlights the challenges in this task and the advantage of the HelmSim.

## 4.2 REAL-WORLD DATA

**Sea Temperature with unknown boundary** Here we use sea water potential temperature in the reanalysis ocean data (MDS) provided by ECMWF. For experiments, we use cropped $64 \times 64$ temperature data in Atlantic, Indian, and South Pacific for training and validation, to be more exact, from 2000 to 2018 for training with 170,249 sequences and from 2019 to 2020 for validation with 17,758 sequences. Additionally, we use the sea temperature in North Pacific from 2000 to 2020 for testing, including 65,286 sequences. This task is to predict the fu-

Table 3: Comparison on Sea Temperature, both MSE and relative L2 are reported.

| Models | Relative L2 | MSE |
|---|---|---|
| DARTS (Ruzanski et al., 2011) | 0.3308 | 0.1094 |
| U-Net (Ronneberger et al., 2015) | 0.1735 | 0.0379 |
| FNO (Li et al., 2021) | 0.1935 | 0.0456 |
| MWT (Gupta et al., 2021) | 0.2075 | 0.0506 |
| U-NO (Rahman et al., 2023) | 0.1969 | 0.0472 |
| LSM (Wu et al., 2023) | 0.1759 | 0.0389 |
| HelmSim (Ours) | **0.1704** | **0.0368** |

ture 10 frames based on the past 10 observations, corresponding to predicting 10 future days based on 10 past days. Since there exists the region shift between training and test sets, this benchmark not only requires the model to capture complex dynamics in ocean but also maintain good generality.

The results in Table 3 demonstrate that HelmSim can handle real-world data well and outperform all baselines. It is worth noting that the test set is collected from different regions with respect to the training and validation sets. These results also verify the generality and transferability of HelmSim.

**Spreading Ink with known boundary** This benchmark consists of three videos collected by Deng et al., each involving 150, 189 and 139 successive frames respectively. Following the experiment setting in Vortex (2023), we split the training and test sets in chronological order by the ratio of 2:1 for each video. Given all the training parts, the goal is to predict all the testing frames at once. For example, for the first video, we need to train our model on the first 100 frames and directly adopt this model to predict the future 50 frames. Since the prediction horizon is much longer than other tasks, this problem poses special challenges in handling accumulative errors.

The quantitative results are listed in Table 4. HelmSim still performs well in the long-term forecasting task. In addition to the relative L2 and MSE, it also consistently achieves the lowest VGG perceptual loss, implying that the prediction results of HelmSim can maintain the realistic texture and intuitive physics. As for showcases in Figure 8, we find that HelmSim can precisely capture the

Table 4: Model comparison on Spreading Ink. Perceptual loss, Relative L2 and MSE are reported.

| Model | Video1 | Video2 | Video3 |
|---|---|---|---|
| U-Net (Ronneberger et al., 2015) | 1.500 / 0.1544 / 0.0080 | 3.982 / 0.4780 / 0.0330 | 5.307 / **0.1535** / **0.0119** |
| FNO (Li et al., 2021) | 2.023 / 0.1709 / 0.0097 | 4.732 / 0.4864 / 0.0342 | 5.531 / 0.1756 / 0.0156 |
| U-NO (Rahman et al., 2023) | 4.210 / 0.1792 / 0.0106 | 6.204 / 0.5312 / 0.0408 | 6.397 / 0.1810 / 0.0166 |
| Vortex (Deng et al., 2023) | 1.704 / 0.1580 / 0.0083 | 4.171 / 0.4150 / 0.0249 | 5.973 / 0.1718 / 0.0150 |
| LSM (Wu et al., 2023) | 1.666 / 0.1592 / 0.0084 | 4.167 / 0.4890 / 0.0346 | 5.448 / 0.1611 / 0.0132 |
| HelmSim (Ours) | **1.464 / 0.1399 / 0.0065** | **3.296 / 0.3565 / 0.0184** | **5.208** / 0.1584 / 0.0127 |

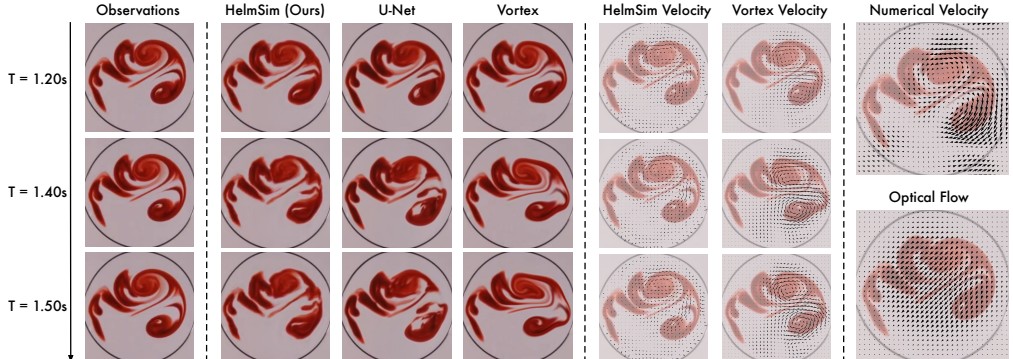

Figure 8: Showcases of prediction results and learned velocity fields on the Spreading Ink dataset.

diffusion of ink. Even for the future 50 frames (T=1.50s), HelmSim still performs well in capturing the hollow position and surpasses numerical methods, optical flow and Vortex in learning velocity.

### 4.3 MODEL ANALYSIS

**Efficiency analysis**    To evaluate model practicability, we also provide efficiency analysis in Figure 9. In comparison with the second-best model U-NO, HelmSim presents a favorable trade-off between efficiency and performance. Specially, HelmSim surpasses U-NO by 12.1% in relative L2 with comparable running time. See Appendix C for full results and comparisons under aligned model size.

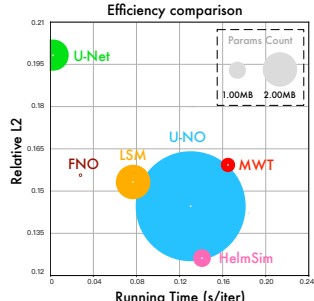

Figure 9: Efficiency comparison. Running time is evaluated on the $64 \times 64$ Naiver-Stokes.

**Ablations**    To highlight the advantages of learning Helmholtz dynamics, we compare HelmSim with two variants: directly learning velocity field and removing the boundary condition design in Helm-Sim. For clarity, we provide both prediction results and learned velocity fields in Figure 10. We can observe that compared to learning Helmholtz dynamics, directly estimating the velocity field will overwhelm the model from capturing complex fluid interactions. Without boundary conditions, learned velocities are perpendicular to the boundary, leading to discontinuous predictions. See Appendix B for quantitative comparisons.

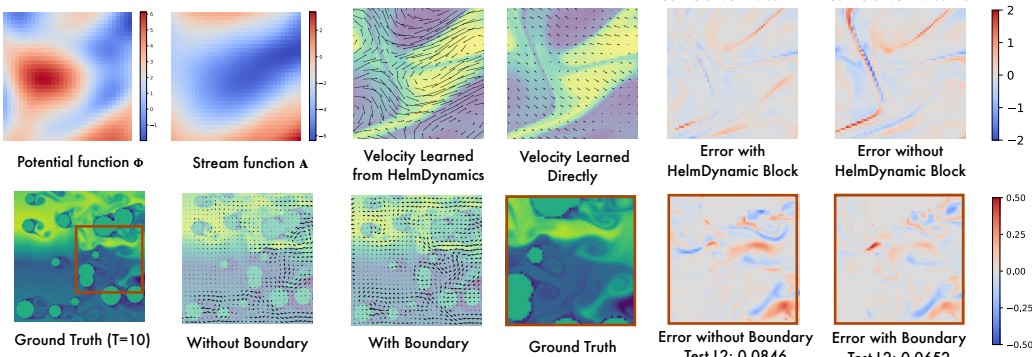

Figure 10: Velocity field and error comparison between learning by HelmDynamic Block and learning directly.

## 5    CONCLUSIONS AND FUTURE WORK

In this paper, we present the HelmSim model towards an accurate and interpretable fluid simulator. Instead of directly learning the velocity field, we propose to learn the Helmholtz dynamics, which casts the intricate dynamics of fluid into inherent physics quantities. With HelmDynamic blocks and Multiscale Integration Network, HelmSim can precisely estimate the potential and stream functions for Helmholtz dynamics, which empowers the prediction process with physical interpretability. HelmSim achieves the consistent state-of-the-art in both simulated and real-world datasets, even for scenarios with complex boundaries. In the future, we will extend HelmSim to 3D fluid simulation and verify its modeling capability in more challenging tasks.

## 6 ETHICS STATEMENT

Our work only focuses on the scientific problem, so there is no potential ethical risk.

## 7 REPRODUCIBILITY STATEMENT

In the main text, we have strictly formalized the model architecture with equations. All the implementation details are included in Appendix A, including dataset descriptions, metrics, model and experiment configurations. The code will be made public once the paper is accepted.

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

Table 5: A summary of experiment datasets. Note that the Spreading Ink dataset is different from other benchmarks, which only contains three video sequences. We strictly follow the Vortex (Deng et al., 2023) to split the video for training, validation and test. For example, in the training phase of video 1, we use the first 70 frames for training and the subsequent 30 frames for validation. As for the test, we use the first 100 frames as input and predict the following 50 frames.

| Dataset | (Input, predict length) | (Training, validation, test) | Observed state | Reynold numbers |
|---|---|---|---|---|
| Navier-Stokes | (10,10) | (1000,200,200) | Vorticity | $\sim 10^4$ |
| Bounded N-S | (10,10) | (1000,200,200) | Grayscale | $\sim 300$ |
| Sea Temperature | (10,10) | (170249, 17758, 65286) | Temperature | Unknown |
| Spreading Ink video 1 | (100, 50) | One video sequence | RGB Image | Unknown |
| Spreading Ink video 2 | (126, 63) | One video sequence | RGB Image | Unknown |
| Spreading Ink video 3 | (93, 46) | One video sequence | RGB Image | Unknown |

## A  IMPLEMENTATION DETAILS

### A.1  DATASET

We summarize the experiment datasets in Table 5. More details can be found in the following.

**Navier-Stokes**  Navier-Stokes equations describe the motion of viscous incompressible field. In this paper, we follow (Li et al., 2021) and generate fluid on a 2D torus with the following equation:

$$\frac{\partial \mathbf{u}}{\partial t} + (\mathbf{u} \cdot \nabla)\mathbf{u} - \nu\nabla^2\mathbf{u} = -\frac{1}{\rho}\nabla p + \mathbf{g}$$
$$\nabla \cdot \mathbf{u} = 0 \tag{9}$$
$$\nabla \times \mathbf{u}(x,0) = \omega_0(x), x \in (0,1)^2,$$

where $\mathbf{u} \in \mathbb{R}^2$ represents the velocity field, $p$ denotes the pressure, and $\rho$ is the fluid density, which we assumed to be constant in the incompressible fluid field. $\nu \in \mathbb{R}_+$ is the kinematic viscosity representing the intrinsic nature of the fluid, which is assumed to be constant. $\mathbf{g} \in \mathbb{R}^2$ represents the summation of all the external forces applied on the fluid field. Vorticity is calculated by the velocity field, $\omega = \nabla \times \mathbf{u}$. At time zero, the initial vorticity field $\omega_0$ is given. The goal is to predict the following vorticity fields from given observations.

We randomly sampled the initial vorticity $w_0$ on a two-dimensional unit torus from a Gaussian distribution, and solved the equation with a numerical method to obtain the future velocity field. After generating the fluid field with $256 \times 256$ spatial resolution and $10^{-4}$ second temporal resolution, we downsampled it to a sequence of 1 second per frame and corresponding spatial resolution. Thus, each sequence consists of 20 frames with a total duration of 20 seconds. We fixed the viscosity $\nu = 10^{-5}$ for all three sub-datasets of different resolutions.

**Bounded N-S**  Suppose a $512 \times 512$ sized two-dimensional space with top and bottom as boundaries, with free space outside the image. We let a randomly colored fluid flow from left to right. To test the model performance under scenarios with complex boundaries, we randomly sampled fifteen circles of different sizes as obstacles and uniformly placed them in the $512 \times 512$ space. Then we used Taichi (Hu et al., 2019) as a simulator engine to generate fluid within top-down boundaries with a numerical fluid solver for the advection equation (Baukal Jr et al., 2000), and generated a sufficiently long sequence with one initial source condition.

Towards the flow field dataset, after the colored fluid field spreads over the space from left to right, we sample frames in the frequency of 60 steps and add the sampled frames into the dataset. To remove the noise from chromatic aberration, we transform all the samples into grayscale. Then we downsample the image to $128 \times 128$, and split the long sequence into disjoint equal-length subsequences. After randomly dividing them into train, validation, and test sets, we finally obtained the training, validation, and test set, which contains 1000, 200, and 200 sequences, respectively.

**Sea Temperature**  As we stated in Section 4.2, we downloaded daily mean sea water potential temperature from the reanalysis ocean data (MDS) provided by ECMWF. Then we selected several areas from both the northern and southern hemispheres, specifically sub-regions of the Atlantic,

Indian, and Pacific Oceans, and then cropped them into $64 \times 64$ patches. For each $64 \times 64$ cropped area, we normalize it to ensure the observations are in a standard distribution, which can make the task free from the noises of sudden change and observation errors, and mainly focus on the dynamics modeling. The task is to predict 10 frames based on the past 10 frames.

**Spreading Ink**  The dataset consists of three open source short videos from Deng et al. (2023). We take each video as a subset and split it into training and test sets along the temporal dimension. This means that the model is trained on the past observations in the training set and is required to predict all the latest fluid in the test set at once, namely the long-term forecasting task.

## A.2  IMPLEMENTATIONS

In this section, we describe the design in incorporating boundary conditions and the aggregation operation of the multiscale integration network.

**Boundary Conditions**  Here we detail the implementation of incorporating boundary conditions as a supplementary of Eq. 4. For given boundary conditions $\mathbb{S}$ and the position $\mathbf{r}$, we calculate the correlation on the intersection between boundary $\mathbb{S}$ and $\mathbf{r}$ neighbour $\mathbb{V}_{\mathbf{r}}$. Concretely, we multiply the boundary mask $\mathbb{1}_{\mathbb{S}}$ to embedded neighbour feature $\widehat{\mathbf{x}}_{T-1}(\mathbf{r}')$, that is,

$$\mathbb{1}_{\mathbb{S}}(\mathbf{r}') \langle \widehat{\mathbf{x}}_T(\mathbf{r}), \widehat{\mathbf{x}}_{T-1}(\mathbf{r}') \rangle = \langle \widehat{\mathbf{x}}_T(\mathbf{r}), \mathbb{1}_{\mathbb{S}}(\mathbf{r}')(\widehat{\mathbf{x}}_{T-1}(\mathbf{r}')) \rangle, \ \mathbf{r}' \in \mathbb{V}. \tag{10}$$

This will preserve the number of neighbour correlation channels, and for $\mathbf{r}' \notin \mathbb{S}$, the correlation values will be set to be zero.

**Aggregation Operation**  Given learned deep representations of prediction $\widehat{\mathbf{x}}^l_{(T+1)}, \widehat{\mathbf{x}}^{l+1}_{(T+1)}$ at the $(l+1)$-th and $l$-th scales, the aggregation operation integrates information between different scales, which can be formalized as follows:

$$\widehat{\mathbf{x}}^l_{(T+1)} = \text{Conv}\left(\text{Concat}\left[\left(\text{Upsample}\left(\widehat{\mathbf{x}}^{l+1}_{(T+1)}\right)\right), \widehat{\mathbf{x}}^l_{(T+1)}\right]\right), \ l \text{ from } (L-1) \text{ to } 1,$$

where we use bilinear interpolation for the operator $\text{Upsample}(\cdot)$.

## A.3  METRICS AND STANDARD DEVIATIONS

In all four datasets, we report the mean value of relative L2 of three repeated experiments with different random seeds as a main metric. Experimentally, the standard deviations of relative L2 are smaller than 0.001 for Navier-Stokes, Bounded N-S and Sea temperature and smaller than 0.003 for Spreading Ink. For scientific rigor, we keep four decimal places for all results. For the Sea Temperature dataset, we report the MSE loss following the common practice in meteorological forecasting. For the Spreading Ink dataset, we used VGG Perceptual Loss (Johnson et al., 2016) to measure the realism of the generated fluids. Given $n$ step predictions $\{\widehat{\mathbf{x}}_i\}_{i=1,\cdots,n}$ and corresponding ground truth $\{\mathbf{x}_i\}_{i=1,\cdots,n}, \widehat{\mathbf{x}}_i, \mathbf{x}_i \in \mathbb{R}^{H \times W}$, the above-mentioned metrics can be calculated as follows:

$$\text{MSE} = \frac{1}{n}\sum_{i=1}^{n}\frac{1}{H \times W}\|\mathbf{x}_i - \widehat{\mathbf{x}}_i\|_2^2, \ \text{Relative L2 Loss} = \frac{\sqrt{\sum_{i=1}^{n}\|\mathbf{x}_i - \widehat{\mathbf{x}}_i\|_2^2}}{\sqrt{\sum_{i=1}^{n}\|\mathbf{x}_i\|_2^2}}.$$

Especially, for Spreading Ink with given boundary conditions $\mathbb{S}$, we only compute the loss function of the area inside the boundary. Suppose that $\mathbb{D}$ represents the area inside the container, and the above-mentioned metrics can be calculated as follows:

$$\text{MSE} = \frac{1}{n}\sum_{i=1}^{n}\frac{1}{|\mathbb{D}|}\sum_{(j,k)\in\mathbb{D}}(\mathbf{x}_{ijk} - \widehat{\mathbf{x}}_{ijk})^2, \ \text{Relative L2 Loss} = \frac{\sqrt{\sum_{i=1}^{n}\sum_{(j,k)\in\mathbb{D}}(\mathbf{x}_{ijk} - \widehat{\mathbf{x}}_{ijk})^2}}{\sqrt{\sum_{i=1}^{n}\sum_{(j,k)\in\mathbb{D}}\mathbf{x}_{ijk}^2}},$$

where $\mathbf{x}_{ijk}$ represents the value at position $(j,k)$ of $i$−th frame, and $|\mathbb{D}|$ represents the number of grid points in $\mathbb{D}$.

## A.4  MODEL AND EXPERIMENT CONFIGURATIONS

All the experiments are implemented in PyTorch(Paszke et al., 2019), and conducted on a single NVIDIA A100 40GB GPU. We repeat all the experiments three times with random seeds selected from 0 to 1000 and report the average results. We train the model with Adam optimizer (Kingma & Ba, 2015) for all baselines. See Table 6 for details.

Here, we present the detailed model configurations for HelmSim. Since fluid in different resolutions will present distinct dynamics, we increase the number of scales for larger inputs, as presented in Table 7. For the Multiscale Integration Network, we follow the conventional design in U-Net (Ronneberger et al., 2015) to downsample, upsample and aggregate multiscale features.

Table 6: Experiment configurations in HelmSim for different benchmarks.

| Benchmark | Learning Rate | Batch Size |
|---|---|---|
| Navier-Stokes | $5 \times 10^{-5}$ | 10 |
| Bounded N-S | $5 \times 10^{-5}$ | 5 |
| Sea Temperature | $5 \times 10^{-5}$ | 10 |
| Spreading Ink | $5 \times 10^{-5}$ | 5 |

Table 7: Hyperparameter configurations of HelmSim for different resolutions.

| Input Resolutions | Hyperparameters | Values |
|---|---|---|
| $64 \times 64$ | Number of scales $L$ | 3 |
| | Number of heads $M$ | 4 |
| | Channels of deep representations $\{d_{\mathrm{model}}^1, \cdots, d_{\mathrm{model}}^L\}$ | $\{64, 128, 128\}$ |
| | Number of neighbours to calculate spatiotemporal correlations $|\mathbb{V}_{\mathbf{r}}|$ | 81 |
| $128 \times 128$ | Number of scales $L$ | 4 |
| $256 \times 256$ | Number of heads $M$ | 4 |
| | Channels of deep representations $\{d_{\mathrm{model}}^1, \cdots, d_{\mathrm{model}}^L\}$ | $\{128, 256, 512, 512\}$ |
| | Number of neighbours to calculate spatiotemporal correlations $|\mathbb{V}_{\mathbf{r}}|$ | 81 |

## B  ABLATION STUDY

We present comprehensive ablations here, including the sensitivity analysis to every hyperparameter, key design in every component of HelmDynamics (stream, potential functions, and boundary conditions), multiscale design in integration.

### B.1  HYPERPARAMETER SENSITIVITY

We include a summary of hyperparameter experiments in Table 8 and conduct detailed ablations in the quantitive aspect as a supplement to Figure 10 of the main text to verify the effect of learning HelmDynamics and considering boundary conditions.

**Order of Runge-Kutta for temporal integration**   Runge-Kutta methods are commonly used for iteratively solving PDEs. The higher the number of orders, the more accurate results it will obtain but it will also consume more computation time. In HelmSim, the prediction results also rely on the accuracy of the velocity obtained by HelmDynamic blocks. According to our experiments, the second-order Runge-Kutta method is already sufficient for temporal integration. Thus, we choose the second-order Runge-Kutta for integration to trade off performance and efficiency.

**Number of neighbours in correlation calculation**   Larger regional area will provide more information for spatiotemporal correlation calculation. In this paper, we choose $|\mathbb{V}_r|$ as $9 \times 9$.

**Number of heads**   Adding heads is a convention to augment model capacity Vaswani et al. (2017). In this paper, adding heads also means more operations in conducting integration. We set $M$ as 4 for a good balance of running time and performance.

Table 8: Model performances on Navier-Stokes Dataset of $64 \times 64$ resolution with different selections for order of Runge-Kutta, number of neighbours in correlations, number of heads and number of scales. The red marked hyperparameter represents the final configuration of HelmSim.

| **Order of Runge-Kutta** | 1 | 2 | 3 | 4 |
|---|---|---|---|---|
| Relative L2 | 0.1298 | **0.1261** | 0.1268 | 0.1278 |
| Training time (s / epoch) | 80.04 | 81.20 | 88.30 | 90.49 |
| **Number of neighbours in correlation $\|\mathbb{V}_r\|$** | 3×3 | 5×5 | 7×7 | 9×9 |
| Parameter number | 9,825,421 | 9,848,461 | 9,883,021 | 9,929,101 |
| Relative L2 | 0.1337 | 0.1273 | 0.1272 | **0.1261** |
| **Number of heads $M$** | 1 | 4 | 8 | 16 |
| Parameter number | 11,063,245 | 9,929,101 | 9,812,653 | 9,762,205 |
| Relative L2 | 0.1344 | 0.1261 | 0.1279 | **0.1249** |
| Training time (s / epoch) | 59.69 | 81.20 | 120.86 | 171.97 |
| **Number of scales $L$** | 2 | 3 | 4 | 5 |
| Parameter number | 9,283,977 | 9,929,101 | 15,906,193 | 29,820,309 |
| Relative L2 | 0.1514 | **0.1261** | 0.1361 | 0.1330 |
| Training time (s / epoch) | 64.43 | 81.20 | 99.83 | 120.06 |

Table 9: Ablations on dynamics learning in $64 \times 64$ Navier-Stokes Dataset.

| Metrics | Multihead version | | Single head version | |
|---|---|---|---|---|
| | Velocity | HelmDynamics | Velocity | HelmDynamics |
| Relative L2 | 0.1412 | **0.1261** | 0.1461 | **0.1344** |
| GPU memory (GB) | 14.86 | 16.30 | 13.02 | 14.41 |
| Training Time (s / epoch) | 72.18 | 80.20 | 48.25 | 61.22 |

**Number of scales** This hyperparameter is highly related to the nature of fluid. Considering both model efficiency and fluid dynamics, we choose $L$ as 3 for $64 \times 64$ inputs and 4 for larger inputs.

## B.2 LEARNING HELMDYNAMICS OR DIRECTLY LEARNING VELOCITY

As we emphasized in the main text, directly learning the superficial velocity will overwhelm the model. As presented in Table 9, without Helmholtz dynamics, the performance decreases from 0.1261 to 0.1412, demonstrating the effectiveness of our proposed Helmholtz dynamics. In addition, the calculation of HelmDynamics only brings marginal extra computation costs.

## B.3 USING OR OMITTING BOUNDARY CONDITIONS

As shown in Table 10, without input boundary condition, the performance drops seriously, indicating the necessity of our design in HelmDyanmic. It is also notable that as a flexible module, it is quite convenient to incorporate boundary conditions into the HelmDyanmic block, which is also a unique advantage of our model in comparing with others.

## B.4 LEARNING HELMDYNAMICS IN MULTIPLE SCALES

As presented in Eq. 8, we ensemble the learned HelmDynamics in multiple scales. Here we also provide ablations on just employing HelmDynamics in one single scale in Table 11. We can find that our multiscale design can facilitate the dynamics modeling.

Table 10: Ablations on boundary conditions in the Bounded N-S dataset.

| Metrics | Omitting Boundary Conditions | Using Boundary Conditions |
|---|---|---|
| Relative L2 | 0.0846 | **0.0652** |
| GPU memory (GB) | 26.98 | 29.48 |
| Training Time (s / epoch) | 226.20 | 267.63 |

Table 11: Ablations on learning HelmDynamics in multiple or single scales.

| Metrics | Multiple Scales | Single Scale (Bottom) | Single Scale (Top) |
|---|---|---|---|
| Relative L2 | **0.1261** | 0.1441 | 0.1798 |
| GPU memory (GB) | 16.30 | 8.80 | 11.68 |
| Training Time (s / epoch) | 80.20 | 30.69 | 45.32 |

## B.5 Are Both Potential and Stream Functions Effective?

As presented in Table 12, only learning potential function or stream function will cause a decrease in the final performance, demonstrating the effectiveness of both components.

## C Efficiency Comparison

In the main text, we have plotted the efficiency comparison. Here, we detail the quantitive results in Table 13 as a supplement.

**Align model size**    Note that all of these baselines are reproduced from their official configurations in their paper, which may result in an unbalanced model size problem. To ensure a fair comparison, we also enlarge the parameter of FNO and compare it with HelmSim. See Table 14 for the results. It is observed that, even enlarging the FNO to a size comparable to HelmSim, it still performs worse.

## D Additional Results

**Align baselines in all benchmarks**    As we stated in Section 4, some of the baselines are not suitable for part of the benchmarks, specifically Vortex (Deng et al., 2023), DARTS Ruzanski et al. (2011), PWC-Net with fluid Refinement (Sun et al., 2018) and MWT (Gupta et al., 2021), which means their performance will degenerate seriously or the running time is extremely slow if we stiffly apply them to all benchmarks. Specifically, due to the special design for temporal information in Vortex, we only compare it in the Spreading Ink dataset in the main text. As for the DARTS, since it is designed for the mass field and not applicable for videos with RGB channels, we do not include it in Spreading Ink dataset. Besides, PWC-Net with fluid Refinement (Zhang et al., 2022) is proposed to learn the optical flow for fluid, which suffers from the accumulative error, making it far inferior to other methods. Thus, we only compare PWC-Net in learning velocity field in the main text.

But, we still provide the missing experiments in Table 15 to ensure transparency.

- Vortex (Deng et al., 2023) models multiple vortex trajectories as a function of time. Since different video sequences have inherently different vortex trajectories, we need to re-train Vortex to fit every video sequence. However, the other three benchmarks except Spreading Ink, have more than 1000 different video sequences. It means that we need to train 1000+ Vortex models for these benchmarks, which is unacceptable. But we still implement this experiment, where we train one vortex model on one single video sequence and generalize it to others.

- Due to the slow movement of the spreading ink dataset, DARTS Ruzanski et al. (2011) showed outstanding quantitive results. However, it fails to predict the correct future in the other three datasets. Moreover, DARTS method solves the least squares problem in the

Table 12: Ablations on learning HelmDynamics, single potential or stream function in $64 \times 64$ Navier-Stokes Dataset.

| Metrics | HelmDynamics | Only potential function | Only stream function |
|---|---|---|---|
| Relative L2 | **0.1261** | 0.1460 | 0.1305 |
| GPU memory (GB) | 16.30 | 16.29 | 16.30 |
| Training Time (s / epoch) | 80.20 | 79.57 | 79.60 |

Table 13: Efficiency comparison between six deep models on Naiver-Stokes $64 \times 64$ dataset, where running time are measured with batch size as 1.

| Models | HelmSim | U-Net | FNO | MWT | U-NO | LSM |
|---|---|---|---|---|---|---|
| #Parameter | 9,929,101 | 17,312,650 | 1,188,641 | 7,989,593 | 61,157,793 | 19,188,033 |
| Training time (s / epoch) | 80.20 | 46.29 | 18.91 | 90.02 | 103.82 | 44.49 |
| Relative L2 | **0.1261** | 0.1982 | 0.1556 | 0.1586 | 0.1435 | 0.1535 |

frequency domain for every case, which will bring huge computation costs. In particular, the other deep methods predict the whole sequence in less than 0.1 seconds, while DARTS takes more than 10 seconds. Also, the changes in estimated velocity are very slight with the change of time, which leads to incorrect location estimation. And the extrapolation causes blurring in long-term prediction.

- PWC-Net (Sun et al., 2018) only estimates the velocity between adjacent observations. We tried to extrapolate with the estimated velocity field to predict the following frames. But this will lead to severe distortion. To better use the estimated velocity, we fed the velocity with the observation into a U-Net (Ronneberger et al., 2015) and got a better result, which is shown as PWC-UNet in table 15. Although enhanced by estimated velocity from PWC-Net, U-Net is still not as good as HelmSim.

- MWT (Gupta et al., 2021) predict the future frames based on wavelet analysis. It fails in long-term prediction. The prediction on video 3 of Spreading Ink (Figure 11) shows that as the prediction time gets longer, the prediction image stays at the same position and appears weird texture.

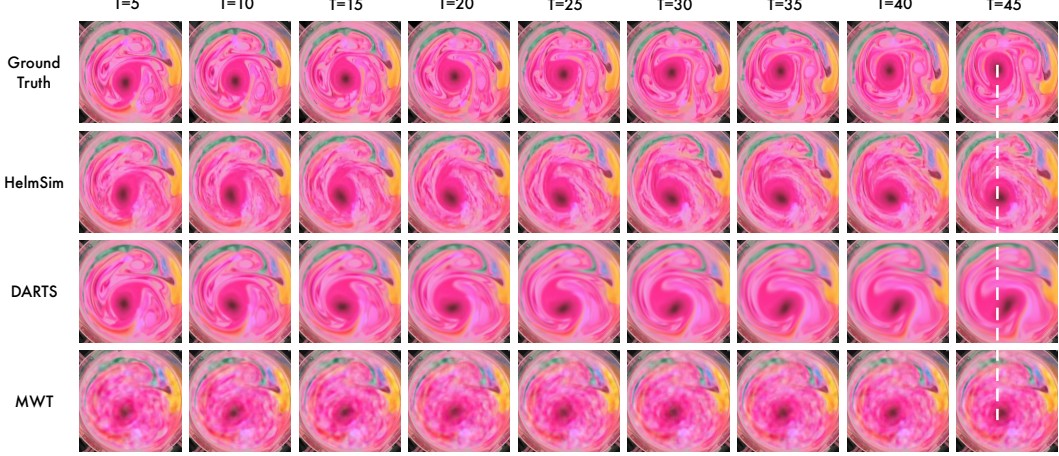

Figure 11: Showcases of HelmSim, DARTS, and MWT on the Spread Ink dataset .

**Performance on turbulence dataset** To better present the model performance on turbulent fluid, we evaluate HelmSim and other baselines on a $64 \times 64$ sized turbulence dataset (Wang et al., 2019), which consists of 6000 sequences for training, 1700 for validation and 2100 for testing. The target

Table 14: Align model size in $64 \times 64$ Navier-Stokes.

| Relative L2 | FNO | FNO (enlarged) | HelmSim |
|---|---|---|---|
| $64 \times 64$ Navier-Stokes | 0.1556 | 0.1524 | **0.1261** |
| $128 \times 128$ Navier-Stokes | 0.1028 | 0.1025 | **0.0807** |
| $256 \times 256$ Navier-Stokes | 0.1645 | 0.1474 | **0.1310** |
| Bounded N-S | 0.1176 | 0.1116 | **0.0652** |
| Sea Temperature | 0.1935 | 0.1958 | **0.1704** |
| Video 1 | 0.1709 | 0.1872 | **0.1399** |
| Video 2 | 0.4864 | 0.5250 | **0.3565** |
| Video 3 | 0.1756 | 0.1676 | **0.1584** |
| Model Parameter | 1,188,641 | 10,633,265 | 9,929,101 |

Table 15: Align baselines in all benchmarks, including DARTS (Ruzanski et al., 2011), adapted version of PWC-Net (Sun et al., 2018), MWT (Gupta et al., 2021), Vortex (Deng et al., 2023). We report Relative L2 for Navier-Stokes dataset and Bounded N-S dataset, MSE and relative L2 for Sea Temperature dataset, and Perceptual loss, Relative L2 and MSE for Spreading Ink.

| | Navier-Stokes | Bounded N-S | Sea Temperature | Spreading Ink (Video 3) |
|---|---|---|---|---|
| DARTS | 0.8046 | 0.1820 | 0.3308 / 0.1094 | 4.940 / 0.1601 / 0.0130 |
| PWC-UNet | 0.1765 | 0.0729 | 0.1805 / 0.0406 | 5.341 / 0.1591 / 0.0128 |
| MWT | 0.1586 | 0.1407 | 0.2075 / 0.0510 | **1.521** / 0.1775 / 0.0160 |
| Vortex | 8.1379 | 1.6259 | 4.9302 / 0.1796 | 5.973 / 0.1718 / 0.0150 |
| HelmSim | **0.1261** | **0.0652** | **0.1704 / 0.0368** | 5.208 / **0.1584 / 0.0127** |

is to predict the following velocity fields given previous observations. To test the best performance on the dataset, we train all the models with an input sequence of 25 timesteps and evaluate for 20. The results are provided in Table 16.

Table 16: Performance on turbulence dataset.

| Turbulence dataset | MSE |
|---|---|
| U-Net (Ronneberger et al., 2015) | 1062.13 |
| TF-Net (Wang et al., 2019) | 1061.78 |
| FNO (Li et al., 2021) | 1187.44 |
| U-NO (Rahman et al., 2023) | 3276.09 |
| LSM (Wu et al., 2023) | 1069.26 |
| HelmSim | **1042.38** |

**Sensitivity to the number of parameters** We also add the sensitivity analysis to the number of parameters on the $64 \times 64$ Navier-Stokes dataset in Table 17. We report the results of changing the channels of deep representations to a half, and twice the original channels. These results show that the original configuration can achieve a favorable balance between performance of efficiency.

## E  DYNAMICS TRACKING

Our design is based on the Euler perspective of fluid. As a supplement, we also provide a Lagrangian perspective comparison to the model predictions. We track one certain fluid particle predicted by HelmSim and other baselines. Technically, we first locate the point with maximum value in a certain area and then keep tracking the maximum-value point in the subsequent frames. As shown in Figure 12, we can find that the closer the trajectory of the point in the predicted image is to its real counter-

Table 17: Ablations on the number of parameters.

| Channels compared to the official configuration | 1/2 | 1 | 2 |
|---|---|---|---|
| Relative L2 | 0.1380 | 0.1261 | **0.1242** |
| GPU memory (GB) | 9.64 | 16.30 | 29.99 |
| Running Time (s / epoch) | 75.10 | 80.20 | 112.13 |
| #Parameter | 2,516,173 | 9,929,101 | 39,446,029 |

part, the better tracking of the point is indicated. The trajectory predicted by HelmSim is the closest to the ground truth, verifying the advantage of HelmSim in dynamics modeling real trajectory.

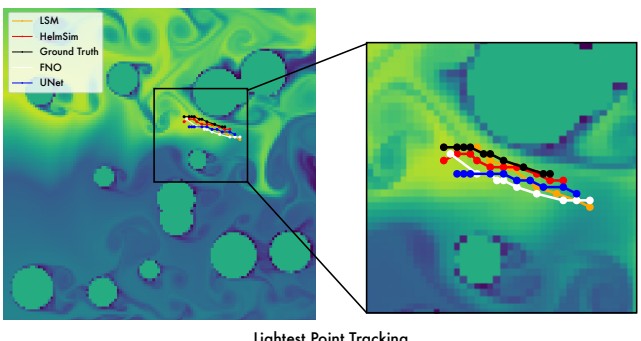

Figure 12: Tracking of the local maxima of Bounded N-S.

## F  FUTURE WORK: EXTEND HELMSIM TO 3D FLUID

Here we present the potential extension of HelmSim to 3D fluid simulation. According to the official formalization of Helmholtz decomposition $\mathbf{F}(\mathbf{r}) = \nabla\Phi(\mathbf{r}) + \nabla \times \mathbf{A}(\mathbf{r}), \mathbf{r} \in \mathbb{V}$, the stream function for a 3D fluid field is a 3D vector $\mathbf{A}(\mathbf{r}) = (\mathbf{A}_x(\mathbf{r}), \mathbf{A}_y(\mathbf{r}), \mathbf{A}_z(\mathbf{r}))$. In 2D cases, the velocity component on the $z$-axis is set to be zero, that is $\mathbf{A}_x(\mathbf{r}) = \mathbf{A}_y(\mathbf{r}) = 0$. To extend HelmSim to 3D fluid simulation, the core is to change the HelmDynamic block into learning $\widehat{\Phi} \in \mathbb{R}^{1 \times D \times H \times W}$ and $\widehat{\mathbf{A}} \in \mathbb{R}^{3 \times D \times H \times W}$, where $D$ is the additional depth dimension of 3D fluid. Then, following the Helmholtz decomposition presented in Eq. 1, we can easily obtain the inferred 3D vector velocity field, thereby enabling HelmSim to achieve the velocity-aware 3D fluid prediction.

## G  MORE SHOWCASES

As a supplement to the main text, we provide more showcases here for comparison.

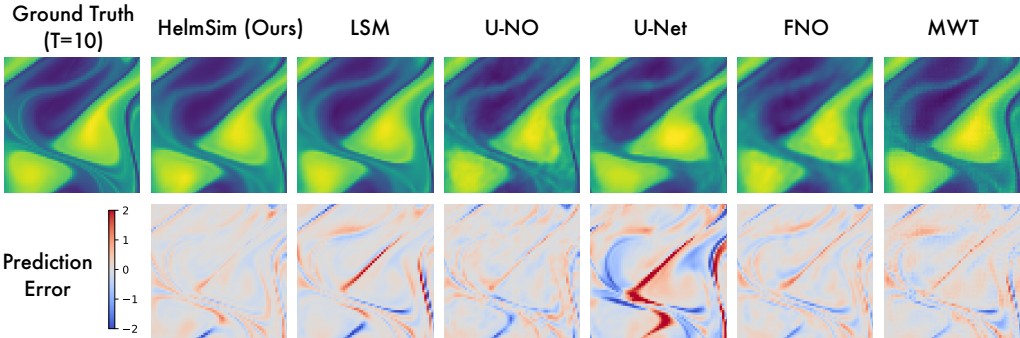

Figure 13: Showcases of the Navier-Stokes Dataset with resolution of $64 \times 64$.

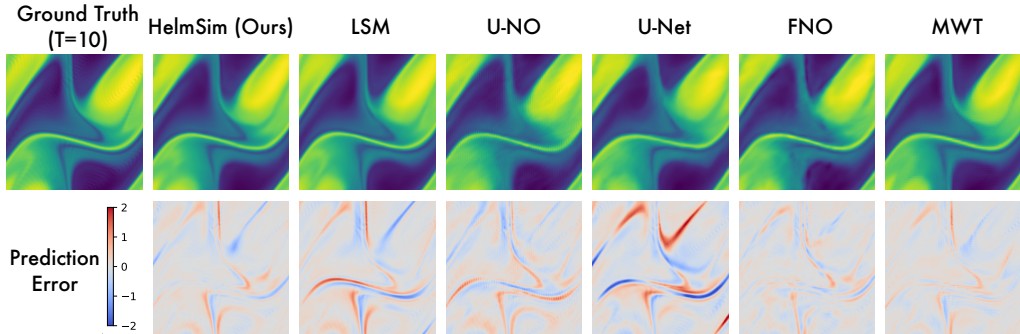

Figure 14: Showcases of the Navier-Stokes Dataset with resolution of $128 \times 128$.

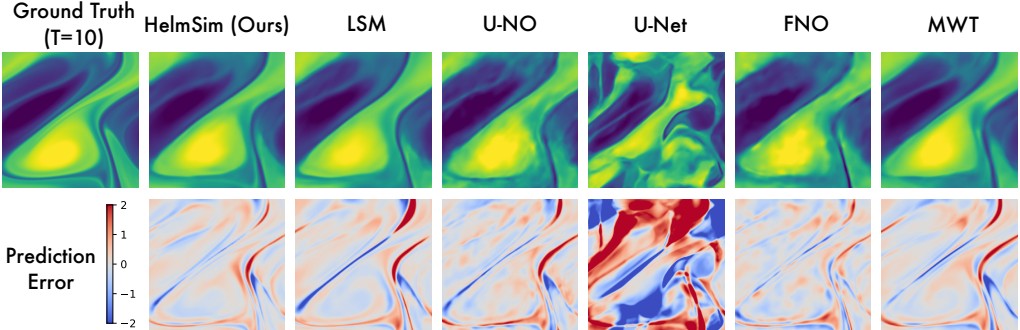

Figure 15: Showcases of the Navier-Stokes Dataset with resolution of $256 \times 256$.

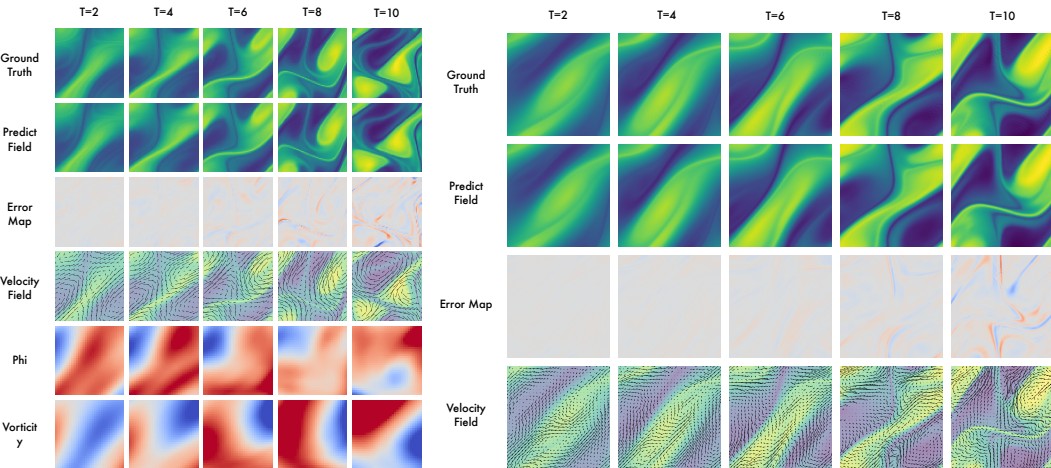

Figure 16: Showcases of HelmSim on Navier-Stokes Dataset with resolution $64 \times 64$ and $128 \times 128$.

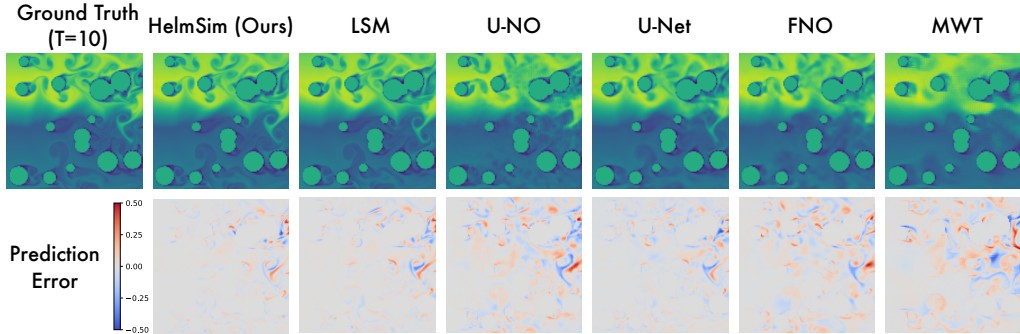

Figure 17: Showcases of the Bounded N-S Dataset.

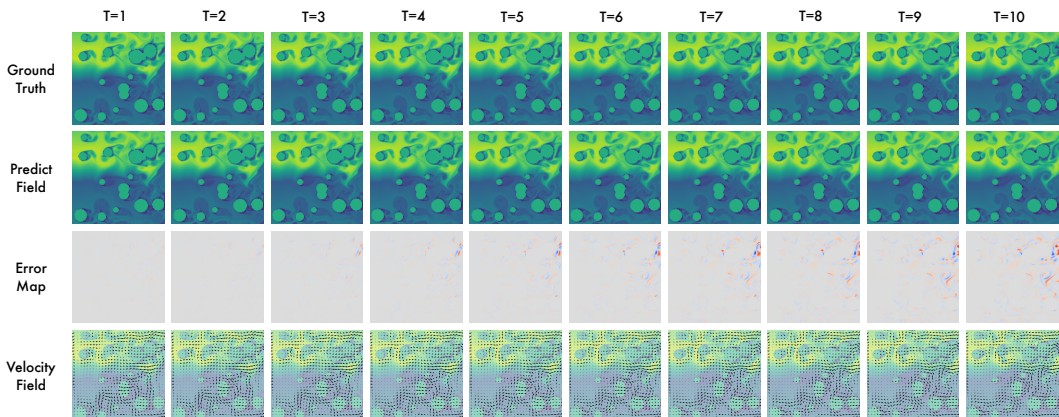

Figure 18: Showcases of HelmSim on the Bounded N-S Dataset.

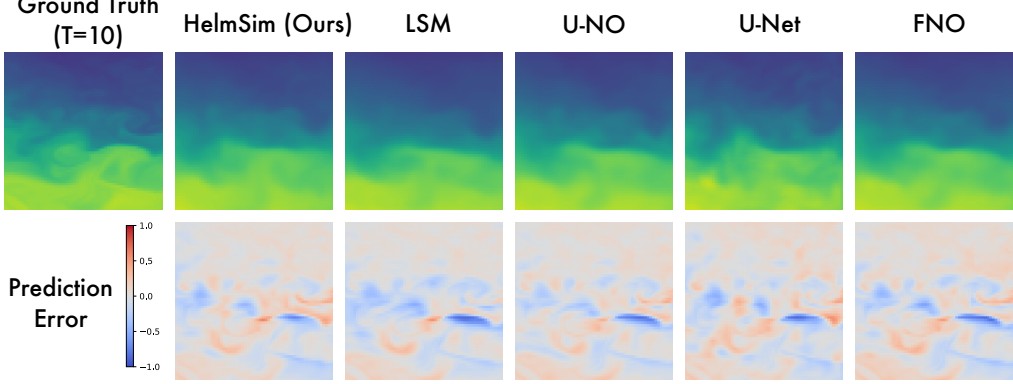

Figure 19: Showcases of the Sea Temperature Dataset.

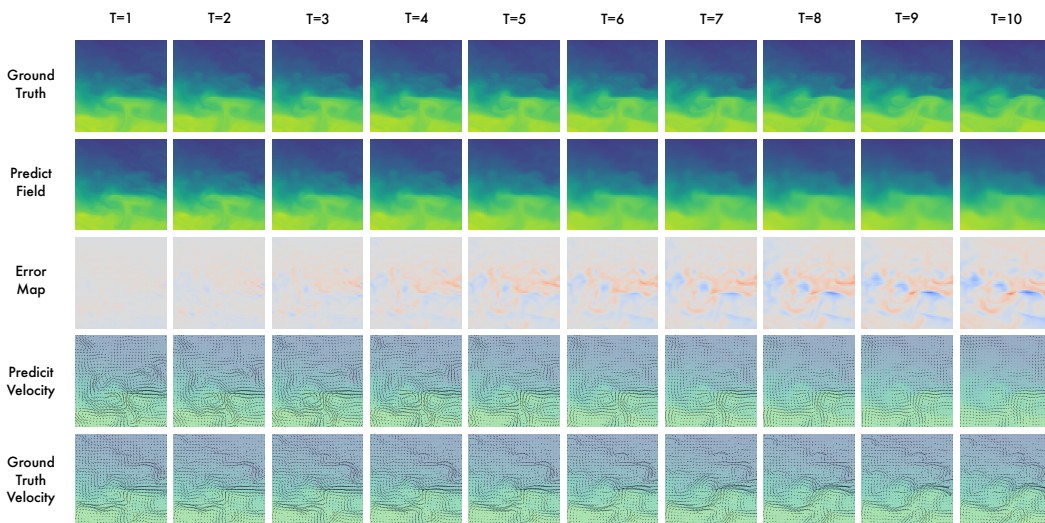

Figure 20: Showcases of HelmSim on the Sea Temperature Dataset.

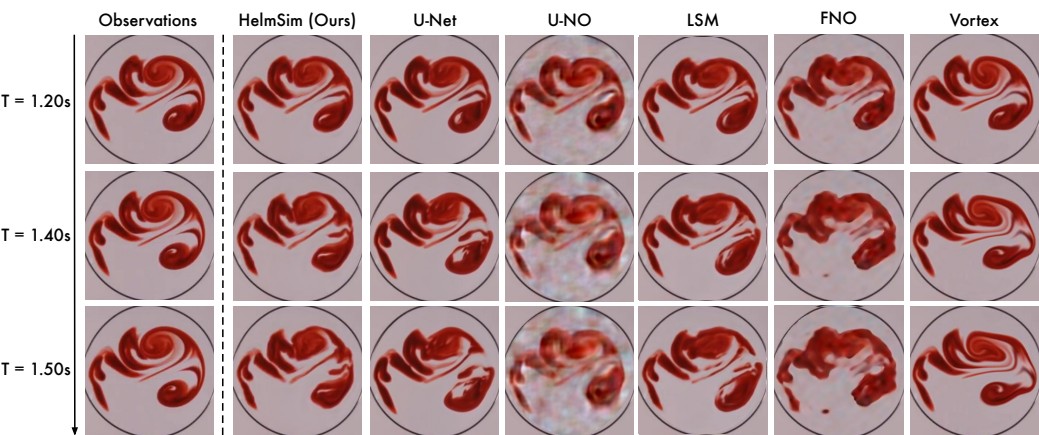

Figure 21: Showcases of the Spreading Ink Dataset (Video 1).

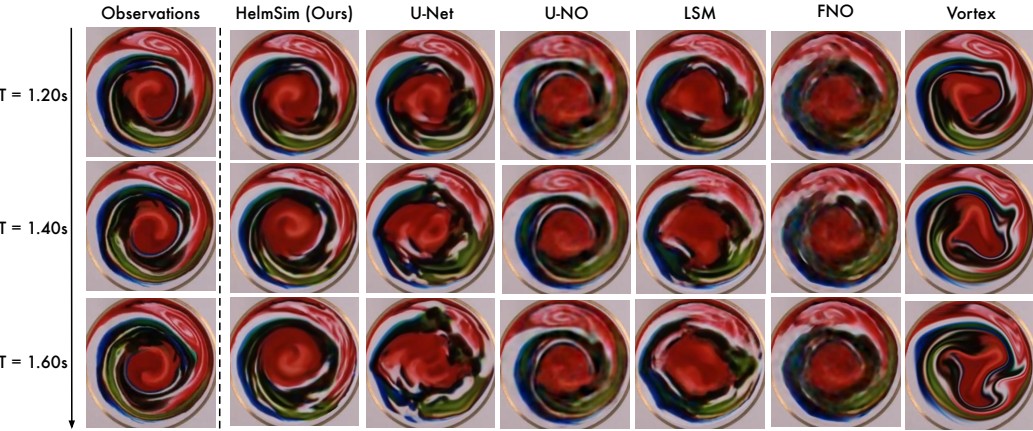

Figure 22: Showcases of the Spreading Ink Dataset (Video 2).

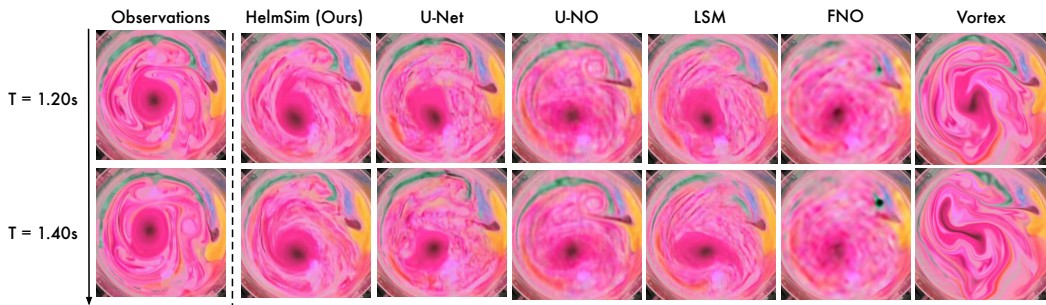

Figure 23: Showcases of the Spreading Ink Dataset (Video 3).