# OpenReview forum: "HelmSim: Learning Helmholtz Dynamics for Interpretable Fluid Simulation"
_ICLR.cc/2024/Conference — Submitted to ICLR 2024_

### Official Review · Reviewer_rfHe · 2023-10-27

**Soundness:** 2 fair
**Presentation:** 2 fair
**Contribution:** 1 poor
**Rating:** 3
**Confidence:** 4

**Summary:**

This paper is motivated by the classic Helmholtz decomposition to decompose fluid flow motions into curl-free and divergence-free parts. This is embedded into a multi-scale architecture and evaluated with multiple two-dimensional examples. Unfortunately, using Helmholtz for learning is a pretty "classic" topic even in the deep learning field, which the paper seems not to be aware of. Hence, the baselines do not take into account numerous approaches that have likewise used curl-based constructions in losses and as neural network layers.

**Strengths:**

The paper targets an important goal, and shows several interesting simulation results. I can encourage the authors to more carefully separate their work from existing papers, and directly show which advantages the approach yields even though I can't recommend accepting the paper in its current form.

**Weaknesses:**

An important point I see with this submission is that it does not seem to be aware of the wide array of papers that use Helmholtz and curl constructions from the fields of deep learning and from the computer graphics. As the paper already targets CG-like flows, and cites several works from this area, I have trouble understanding why the authors have omitted all the classic works from this area.

However, most important of course are directly relevant DL papers, that have used very similar constructs. The classic "Accelerating eulerian fluid simulation with convolutional networks" Tompson et al. ICML 2017 uses it in the loss,  and "Deep fluids: A generative network for parameterized fluid simulations" Kim et al. 2019 use it for inferring flow solutions, very closely related to the paper here.

Many newer papers likewise use curl constructions, e.g. "Curl-Flow: Pointwise Incompressible Velocity Interpolation forGrid-Based Fluids" Chang et al. 2021, and "Discovering Hidden Physics Behind Transport Dynamics" Liu et al. 2021. Also closely related, just last year at ICLR 3D flows were synthesized using a multi-scale architecture with curl: "Learning to Estimate Single-View Volumetric Flow Motions without 3D Supervision"  Franz et al. 2023. None of these papers are cited, or compared to. This is a big omission, and leaves many questions open. Most importantly, whether there are any gains from the proposed approach over these already published works.

In addition, I would recommend citing and acknowledging the classic CG papers: Bridson's curl-noise paper, and derivatives such as "Stream function solver for liquid simulations" Ando et. al 2015. Beyond this, the related work in terms of works from the DL field focusing on fluids also looks like it could be broadened and improved.

Beyond the unclear advantages, I was also surprised that the paper focuses on extremely short prediction horizons. 10 steps do not seem like a state of the art for current DL-based simulators. I think it will be important to target much longer roll-outs in order to keep up with existing works. (I would expect the higher order time integration methods to relatively quickly cause trouble in these cases.)

In general, the appendix is quite sparse w.r.t. additional details. I think future revisions of the submission could also be improved by providing further ablations and details here.

**Questions:**

It is not intuitive to me why the Helmholtz decomposition should be enforced throughout the network, and on multiple scales. In the end, only the final output should adhere to this, and I would expect it to overly constrain the inner latent spaces of a network. Have the authors tried adding only a single layer at the end?

It's crucial for a fair comparison to keep the number of network parameters similar. Why have the authors chosen to vary them so widely? There's a factor of 20x between smallest and largest in Table 9. Along those lines, 10m for a 64x64 flow seem huge. I would recommend to add an ablation for different sizes.

---

Post-rebuttal comments:

I want to thank the authors for the updates and their replies. However, I really cant understand why the authors are trying to argue that their "task" is not the simulation of a fluid, but rather an inverse problem of inferring velocities from some observed state. To quote the abstract "we propose the HelmSim toward an accurate and interpretable simulator for fluid" , and the main body continues like this. If the goal of the paper is really _not_ the simulation of fluids, but some derived inverse task, I would recommend to reflect this in introduction and the presentation of the method and rewrite them accordingly. However, if the goal is simulating fluids after all, I think it's important how the method fares for actual simulation tasks using velocities and central input output quantity (instead of only focusing on velocity estimation tasks).

I do evaluate this paper based on its currently stated objectives, i.e. simulating fluids, and for this 10 steps are clearly not sufficient. Current learning-based methods typically focus on hundreds or thousands of steps. As such, I will keep my score. I don't think this paper should be published at ICLR in its current form, but would require some fundamental revisions that should be carefully checked by reviewers.

Table 12 also contains the stream only / curl only versions. I think future versions of the paper should make clear that "curl only" is widely used in the DL community already. Being very close to the full version, this seems to give the majority of the gains.

---

> ### Author Response · Authors · 2023-11-18
> **Response to Reviewer rfHe [Part 1]**
>
> Many thanks to Reviewer rfHe for providing a detailed review and insightful questions.
>
> > **Q1:** Omitted all the classic works from Computer Graphics.
>
> Many thanks for the reviewer recommending these papers to us. **But after carefully reading these papers, we regretly find that none of these papers can be applied to our task. Our paper predicts the future solely based on the observed physics quantities, and the ground truth velocity is unaccessible.**
>
> Considering these papers are partially related to our topic, we add a new paragraph naming as **computer graphic for fluid simulation** in the $\underline{\text{related work  section of revised paper}}$ to discuss these CG-based works. Here are some key points.
>
> | Papers                                                       | Mismatch Point                                               |
> | ------------------------------------------------------------ | ------------------------------------------------------------ |
> | Accelerating Eulerian fluid simulation with convolutional networks | It uses intermediate results from numerical algorithms and is designed for inviscid fluid. |
> | Deep fluids: A generative network for parameterized fluid simulations | It uses ground truth velocity supervision.                   |
> | Curl-Flow: Pointwise Incompressible Velocity Interpolation for Grid-Based Fluids | It requires velocity field input and restricts the velocity field to be divergence-free. Their target is to interpolate, not predict. |
> | Discovering Hidden Physics Behind Transport Dynamics         | It uses ground truth velocity supervision.                   |
> | Learning to Estimate Single-View Volumetric Flow Motions without 3D Supervision | It is not designed for predictive tasks.                     |
> | Stream function solver for liquid simulations                | It requires velocity field observation as input, and cannot model time-variant changes of scalar field. |
>
> > **Q2:** The paper focuses on extremely short prediction horizons.
>
> As we stated in **Q1**, we attempt to predict the future fluid solely based on one observed physical quantity, which may be different from your conventional understanding of the fluid simulation task.
>
> Following the previous paper, we establish our experiment as follows:
>
> | Benchmark       | Task                 | Previous work |
> | --------------- | -------------------- | ------------- |
> | Navier-Stokes   | Input-10-predict-10  | FNO [1]       |
> | Bounded N-S     | Input-10-predict-10  | GNOT [2]      |
> | Sea Temperature | Input-10-predict-10  | Equ-Net [3]   |
> | Spreading Ink   | Input-100-predict-50 | Vortex [4]    |
>
> Thus, we believe that our experiment setting is well-established and can well examine the model's effectiveness.
>
> Reference:
> [1] Li, Zongyi, et al. "Fourier neural operator for parametric partial differential equations." ICLR 2021.
>
> [2] Hao, Zhongkai, et al. "Gnot: A general neural operator transformer for operator learning." ICML 2023.
>
> [3] Wang, Rui, et al. "Incorporating symmetry into deep dynamics models for improved generalization." ICLR 2021.
>
> [4] Deng, Yitong, et al. "Learning Vortex Dynamics for Fluid Inference and Prediction." ICLR 2023.
>
> > **Q3:** Why the Helmholtz decomposition should be enforced throughout the network? Have the authors tried adding only a single layer at the end?
>
> Note that the multiscale property is one of the foundation features of fluid. Thus, we attempt to capture the complex dynamics in multiple scales. Following the reviewer's suggestion, we also experiment with the case that only Helmholtz decomposition to the end layer as follows. We can find that our proposed multiscale learning is essential.
>
> | 64 × 64 Navier-Stokes                                       | Relative L2 |
> | ----------------------------------------------------------- | ----------- |
> | HelmSim (only add Helmholtz decomposition to the end layer) | 0.1798      |
> | HelmSim                                                     | **0.1261**  |

---

> ### Author Response · Authors · 2023-11-18
> **Response to Reviewer rfHe [Part 2]**
>
> > **Q4:** An ablation for comparable parameter sizes of different models.
>
> Firstly, we want to emphasize that all the baselines are compared with their official configurations in their paper, which means FNO is proposed as a lightweight model.
>
> To ensure a fair comparison, we enlarge the FNO parameter and compare it with HelmSim as follows. We can find that, even with sufficient model parameter, FNO still performs worse than HelmSim.
>
> | Model                       | Relative L2 in 64 × 64 Navier-Stokes | Model parameter |
> | --------------------------- | ------------------------------------ | --------------- |
> | FNO (offical configuration) | 0.1556                               | 1,188,641       |
> | FNO (enlarge model)         | 0.1524                               | 10,633,265      |
> | HelmSim                     | **0.1261**                           | 9,929,101       |
>
> > **Q5:** An ablation for HelmSim of different parameter sizes.
>
> We here add results to compare the sensitivity to the number of parameters on the $64\times 64$ Navier-Stokes dataset. Considering GPU memory and input channels, we report the results of changing the channels of deep representations to half or twice the original channels. The results show that the original parameter number is an efficient and accurate setting.
>
> | Channels compared to the original setting | 1/2       | 1         | 2          |
> | ----------------------------------------- | --------- | --------- | ---------- |
> | Relative L2                               | 0.1380    | 0.1261    | 0.1242     |
> | GPU memory (GB)                           | 9.64      | 16.30     | 29.99      |
> | Running Time (s / epoch)                  | 75.10     | 80.20     | 112.13     |
> | #Parameter                                | 2,516,173 | 9,929,101 | 39,446,029 |
>
> **We sincerely thank the reviewer for providing valuable suggestions. However, we think that there might exist some misunderstandings of reviewer in our experiment setting or scope. During the rebuttal period, we made a great effort to clarify our implementation details and ensure a fair and comprehensive comparison. We hope the reviewer can re-examine the scope of our paper.**

---

> ### Author Response · Authors · 2023-11-20
> **Request of Reviewer's attention and feedback**
>
> Dear Reviewer,
>
> We kindly remind you that it has been 2 days since we posted our rebuttal and there are only 3 days left in the reviewer-author discussion period. Please let us know if our response has addressed your concerns.
>
> Following your suggestion, we have answered your concerns and improved the paper in the following aspects:
>
> - We dived into 6 papers you recommended but regretly found that none of these papers can be applied to our task. To clarify the differences in experiment settings, **we add a new paragraph in the related work section to discuss the CG-based works.**
> - We also **clarify the experiment settings** of our paper by listing previous papers that we followed.
> - We added ablation of **different parameter numbers, multiscale HelmDynamic Block** to show our efficient and accurate design.
>
> **All of these results have been included in the $\underline{\text{revised paper highlighted in blue}}$.**
>
> Thanks again for your valuable review. We are looking forward to your reply and are happy to answer any future questions.

---

> ### Author Response · Authors · 2023-11-23
> **The discussion period ending soon**
>
> Dear Reviewer,
>
> Thanks again for your valuable and constructive review, which helps us clarify the relation with CG-based methods and improve the experiment with detailed ablations. All the updates have been included in the $\underline{\text{revised paper highlighted in blue}}$.
>
> **We kindly remind you that the reviewer-author discussion phase will end in a few hours. May we know if our response addresses your main concerns? After that, we will not have a chance to respond to your comments**.
>
> Sincere thanks for your dedication! We are looking forward to your feedback.

---

> ### Author Response · Authors · 2023-11-23
> **We are anticipating your feedback**
>
> Dear Reviewer,
>
> We kindly remind you that **the 12-day reviewer-author discussion only left 1 hour**.
>
> In the rebuttal, we have answered your concerns by clarifying the relation between HelmSim and CG-based methods, providing new ablation studies on model parameter and multiscale design. **All the updates have been included in the $\underline{\text{revised paper}}$, including 8 pages of new results and analysis.**
>
> Many thanks for your dedication in reviewing paper, which helps us a lot in imporving our work. **Would you please kindly let us know if our response addresses your main concerns? We eagerly await your reply.**

---

### Official Review · Reviewer_2nz9 · 2023-10-30

**Soundness:** 2 fair
**Presentation:** 3 good
**Contribution:** 2 fair
**Rating:** 5
**Confidence:** 3

**Summary:**

The paper tackles fluid learning problems specifically, where the velocity field can be decomposed into potential and stream functions through the Helmholtz decomposition. A supervised framework is proposed leveraging this decomposition to ease the learning process of the network on simulated and real data in 2D.

**Strengths:**

The motivation for the problem is well-introduced, and contributions are clear. Visuals are also designed properly to aid understanding the concepts proposed by the paper. An appropriate amount of background is provided as well. The benchmarks show the capability of the models, especially with the inclusion of real-world datasets.

**Weaknesses:**

1. It is unclear what the observed state is at all, "observed fluid" could be velocity, pressure, or density fields. This should be mentioned.
2. The experimental results are reported as single relative L2 errors. Are these compared over networks trained on various random seeds, or how was this single value chosen? Ideally we can see a statistically significant improvement with the proposed model.
3. More detail should be added on the boundary condition inclusion, how is this implemented in an efficient manner?
4. The last sentence on page 4 should have hats on the states $x_T$ and $x_{T-1}$, is that correct?
5. Figure 9 shows an efficiency comparison, is it correct to assume the y-axis performance is equivalent to accuracy of the model? Higher is better? Why would HelmSim be at the lower end of the spectrum then? Especially since it's the second slowest model in runtime.
6. The number of parameters is different between each model, it would make sense to make a fair comparison where the same number of parameters was provided for each model. In this particular case, FNO was provided roughly 9x less parameters to train on. The number of parameters should ideally also be mentioned in the main text.
7. Figure 10 should be presented in a more quantitative manner. The boundary condition errors do not look significantly different, it would help to know what sort of percentage difference on average it has on the overall prediction error.

**Questions:**

1. What would have to be added for this to generalize to 3D? The argument in the paper is that 3D real data is hard to observe. So what about staying in simulation, would the model easily generalize to that?
2. It was unclear until the multiscale integration network section that we were operating with a Lagrangian and not a Eulerian fluid learning setup, since we are learning a velocity field and using this to advect field quantities. Is this interpretation correct?
3. In the experimental results, the authors mention that the blurry predictions from U-Net "impede its practicability". In what application areas will these models be applied?
4. The argument for why directly estimating the velocity will overwhelm the model in the ablation study is not very convincing. Is that not what the other baseline models are already capable of?

---

> ### Author Response · Authors · 2023-11-18
> **Response to Reviewer 2nz9 [Part 1]**
>
> Many thanks to Reviewer 2nz9 for providing the insightful review and comments.
>
> > **Q1:** It is unclear what the observed state is at all.
>
> Following the reviewer's suggestion, we have added the observed state information in $\underline{\text{Appendix A.1 of revised paper}}$.
>
> > **Q2:** Are these compared over networks trained on various random seeds, or how was this single value chosen?
>
> Sorry for this missing information. We repeated all the experiments three times with seeds randomly selected from 0 to 1000. The presented results are the mean value of three repeated experiments.
>
> Experimentally, the standard deviations of relative L2 are smaller than 0.001 for Navier-Stokes, Bounded N-S and Sea temperature and smaller than 0.003 for Spreading Ink. For scientific rigor, we keep four decimal places for all results. We have included this in $\underline{\text{Appendix A.2 and A.3 of revised paper}}$.
>
> > **Q3:** More details about boundary condition inclusion.
>
> As we presented in $\underline{\text{Eq. (4) of original submission}}$, we add the boundary condition to the correlation calculation process. A detailed description is also added in $\underline{\text{Appendix A.2 of revised paper}}$.
>
> > **Q4:** Typo of the last sentence on page 4.
>
> Thanks for the suggestion. We have corrected this typo in the $\underline{\text{revised paper}}$.
>
> > **Q5:** Explanation of Figure 9: Efficiency comparison.
>
> Sorry for the misleading y-axis label. The y-axis is Relative L2, which means that a smaller value refers to better performance. We have rephrased this figure in the $\underline{\text{revised paper}}$.
>
> > **Q6:** FNO with roughly 9x less parameters to train on compared to HelmSim. Compare HelmSim to FNO with similar model size.
>
> Firstly, we want to emphasize that all the baselines are compared with their official configurations in their paper, which means FNO is proposed as a lightweight model.
>
> Following the reviewer's suggestion, we enlarge the FNO parameter and compare it with HelmSim as follows. We can find that, even with sufficient model parameters, FNO still performs worse than HelmSim.
>
> | Model                       | Relative L2 in 64 × 64 Navier-Stokes | Model parameter |
> | --------------------------- | ------------------------------------ | --------------- |
> | FNO (offical configuration) | 0.1556                               | 1,188,641       |
> | FNO (enlarge model)         | 0.1524                               | 10,633,265      |
> | HelmSim                     | **0.1261**                           | 9,929,101       |

---

> > ### Author Response · Authors · 2023-11-18
> > **Response to Reviewer 2nz9 [Part 2]**
> >
> > > **Q7:** Figure 10 should be presented in a more quantitative manner.
> >
> > Actually, as we stated in the ablation paragraph, we have already included the quantative results in $\underline{\text{Table 8 of original submission}}$. For clarity, we also add the quantitive results to the $\underline{\text{Figure 10 of revised paper}}$.
> >
> > > **Q8:** What would have to be added for HelmSim to generalize to 3D?
> >
> > HelmSim can be naturally generalized to 3D fluid by learning 3D potential and stream functions. We have added a new section in $\underline{\text{Appendix F of revised paper}}$ to discuss this.
> >
> > > **Q9:** Whether Lagrangian or Eulerian fluid learning setup do you use?
> >
> > We adopt the Eulerian learning setting. The dynamic tracking visualization is just to measure the model performance from a new persepective.
> >
> > > **Q10:** Blurry predictions from U-Net "impede its practicability". In what application areas will the predictions from HelmSim and other baselines be applied?
> >
> > The experimental benchmarks clearly present the applications. For example, in the Sea Temperature benchmark, temperature prediction can provide guidance to the fishery industry. A blurry prediction will fail to provide instructive information to this application due to the inaccurate results in predicting extreme values.
> >
> > > **Q11:** Why directly estimating the velocity will overwhelm the model in the ablation study is not very convincing.
> >
> > As we presented in $\underline{\text{Figure 1,6,8,10 of original submission}}$, directly learning velocity will make the model fail in learning correct dynamics, especially the divergence-free part. However, the divergence-free part is well covered by the learning stream function in HelmSim.
> >
> > The quantitive results are also provided in $\underline{\text{Table 9 of original submission}}$. We also place the comparison here, where learning Helmholtz Dynamics can boost the performance with a little extra computation costs.
> >
> > | 64 × 64 Navier-Stokes Dataset      | Relative L2 | GPU memory (MiB) | Running Time (s/epoch) |
> > | ---------------------------------- | ----------- | ---------------- | ---------------------- |
> > | Directly learning velocity         | 0.1412      | 14.86            | 72.18                  |
> > | Learning Helmholtz Dynamics (Ours) | **0.1261**  | 16.30            | 80.20                  |

---

> ### Comment · Reviewer_2nz9 · 2023-11-21
>
> Thank you to the authors for the extensive replies to all the reviewers, and for grouping the concerns into specific categories.
>
> Thanks for including so much extra detailed information in the appendix, that will ensure reproducibility and clarity of the paper. Indeed, the comparison with FNO is very much appreciated, and now clearly shows the improvement of the proposed HelmSim approach. The additional section to 3D is perhaps slightly weak, since as the authors mentioned, it is a larger challenge within the community, and it would have been a great addition to see realized within this paper.
>
> On the topic of how directly learning the velocity "overwhelms" the model, the way this could be understood is that we craft a simpler representation where the solution is easier to learn. Which goes into engineering features for neural network representations. In this case it is interpretable, yet it remains unclear if this decomposition-based representation is either the easiest to learn or the most physically interpretable. Where one would rather use dimensionality reduction techniques for the first, and potentially directly learn velocities and pressures for the latter. This is only a small comment to add, and perhaps a bigger problem challenge to tackle in the future.
>
> One concern still remains, and perhaps has been amplified by the extra experiments, that the improvement feels somewhat incremental. Same with learning velocity directly, where it is a slight trade-off of runtime/memory for accuracy. It would be great if the model outperforms all baselines by a small amount, but excels in specific scenarios. This one specific scenario is what is missing to make this an excellent paper as opposed to a good paper.
>
> However, the answers from the authors, in addition to the extensive benchmarks, have proven satisfactory and convinced me to improve my score. Thank you once again for clarifying a lot of the points,

---

> > ### Author Response · Authors · 2023-11-22
> > **Thanks for your response**
> >
> > Many thanks to the reviewer's new response. We appreciate that the reviewer acknowledged our "**extensive replies**".
> >
> > Especially, we would like to further clarify our promotion. Here we highlight some key results, where we can find that HelmSim will bring more than 10% relative promotion over the previous best model in the following two well-established benchmarks.
> >
> > | Relative L2         | 64x64 Navier-Stokes | Boundary N-S |
> > | ------------------- | ------------------- | ------------ |
> > | Previous best model | 0.1435 (U-NO)       | 0.0737 (LSM) |
> > | HelmSim             | 0.1261              | 0.0652       |
> > | Relative Promotion  | 12.1%               | 11.5%        |
> >
> > Besides, as we presented in our ablations, the design for learning Helmholtz Dynamics instead of directly learning velocity decreases the relative L2 from 0.1412 to 0.1261 in the 64x64 Navier-Stokes dataset, leading to a 12% improvement. We believe that such significant promotion is meaningful to real-world applications.
> >
> > Sincerely thanks for your response and dedication. I also noticed that you raised the score to "5: marginally below the acceptance threshold". Given the newly added experiments, clarification, and significant promotion, we do hope you can reexamine the final judgment. If possible, would you please kindly itemize the potential rejection reasons, we are happy to answer any further questions.

---

### Official Review · Reviewer_Mqos · 2023-10-31

**Soundness:** 2 fair
**Presentation:** 2 fair
**Contribution:** 2 fair
**Rating:** 6
**Confidence:** 3

**Summary:**

This paper introduces a model that regularizes the prediction of a neural network for the velocity field or the quantity advected by this velocity in 2D fluid dynamics. The framework decomposes the prediction of the velocity vector field according to the Helmholtz theorem into two separate terms and prediction the dynamics following the numerical scheme in multiple scales. The model achieves better results than non-regularized data-driven models.

**Strengths:**

- The paper addresses an important problem concerning partially-observed dynamical systems, which is a complex and underdressed problem.
- I commend the authors for their extensive testing of their methods across a wide range of datasets, which is a valuable contribution to the research.
- The paper explores an alternative approach to structurally integrate the inherent correlation between $F_x$ and $F_y$ components, drawing from the Helmholtz decomposition theorem to constrain the generation of the velocity field.

**Weaknesses:**

- Source of Improvement and Ablation Study:
  - Given the presence of various complex architectural choices, it's difficult to determine whether the Helmholtz decomposition is the primary source of the observed performance improvement. Notably, the absence of the multi-head mechanism leads to a performance drop (0.1261 -> 0.1344) for the 64x64 Navier-Stokes, which is somewhat comparable to the performance decrease resulting from the ablation of the Helmholtz decomposition (0.1261 -> 0.1412). These results raise questions about the model's overall performance gain compared to the baseline models when the multi-head trick is absent. Additionally, the ablation studies need to be explained more comprehensively with sufficient details, as the current presentation makes it difficult to understand the methodology and outcomes.
  - The paper claims that Vortex (Deng et al., 2023) cannot be tested on other datasets, which seems unusual, as they are the same type of task and data that are disconnected from the choice of dynamics modeling itself. It should be further clarified why Vortex cannot be applied to other datasets.
- Interpretability Claim:
  - The paper's claim about interpretability is not well-explained. If the interpretability claim is based on the model's prediction of an explicit term of velocity, it needs further comparison and a more comprehensive explanation. Does the Helmholtz decomposition significantly improve interpretability compared to baseline models, such as Vortex (Deng et al., 2023)?
  - In Figure 4, it appears that the model predicts incoherent velocity fields around the circle boundary, even with non-zero velocity outside the boundary, while baseline models do not exhibit such artifacts. This weakens the interpretability claim.
- Multiscale modeling:
  - The aggregation operation after "Integration" needs further clarification. Please provide more details in the main paper, and if you refer to other architectures, acknowledge their structure properly.
- Regarding some missing experimental results with cited baselines, it's crucial to include and report all baseline results to ensure transparency, even if the outcomes are considered inferior.
- Minor issues:
  - Ensure proper citation format for baseline models (Authors, Year).
  - Make sure that symbols are well-defined with clear reference to their definitions. For example, in Equation (4), the undefined operator $\mathbb{I}_{\vec r\in\mathbb{S}}$ needs clarification. If it's an indicator function, use standard notation with a proper explanation. "Embed(•)" should be indicated more explicitly.

**Questions:**

- Are there additional insights or reasons for employing multi-head integration beyond the expected capacity and performance improvements? It would be helpful to understand the broader intuitions behind this approach.
- Have the authors attempted to compare the Helmholtz decomposition with a Clifford Layer (Brandstetter et al., 2022) as they both aim to achieve the correlation of different components of the velocity field?

References:
- Johannes Brandstetter et al., Clifford Neural Layers for PDE Modeling,

---

> ### Author Response · Authors · 2023-11-18
> **Response to Reviewer Mqos [Part 1]**
>
> We would like to sincerely thank Reviewer Mqos for providing valuable feedback.
>
> #### **Section 1: Source of Improvement and Ablation Study**
>
> > **Q1:** Whether the Helmholtz decomposition is the primary source of the observed performance improvement?
>
> We added new ablations on with or without Helmholtz Dynamics and with or without multihead, here are the results:
>
> | Relative L2 on 64 × 64 Navier-Stokes Dataset                 | 1 head | 4 head     | Improvement ($1-\frac{\text{1 head}}{\text{4 head}}$) |
> | ------------------------------------------------------------ | ------ | ---------- | ----------------------------------------------------- |
> | Directly learning velocity                                   | 0.1461 | 0.1412     | 3.35%                                                 |
> | Learning Helmholtz Dynamics                                  | 0.1344 | 0.1261     | 6.18%                                                 |
> | Improvement ($1-\frac{\text{Helmholtz dynamics}}{\text{Directly learning velocity}}$) | 8.00%  | **10.70%** |                                                       |
>
> It is obsevered that the improvement brought by learning Helmholtz Dynamics is much more significant than the improvement brought by multihead design.
>
> > **Q2:** The ablation studies need to be explained more comprehensively with sufficient details.
>
> Following the reviewer's suggestion, we have rephrased the $\underline{\text{Appendix B of revised paper}}$ with newly added implementation details for each ablation.
>
> > **Q3:** Why Vortex cannot be applied to other datasets besides Spreading ink dataset?
>
> Vortex models multiple vortex trajectories as a function of time. Since different video sequences have inherently different vortex trajectories, we need to re-train Vortex to fit every video sequence. However, the other three benchmarks, for example, the Navier-Stokes dataset, have more than 1000 different video sequences. It means that we need to train 1000+ Vortex models for these benchmarks, which is unacceptable. But as per your request, we still implement this experiment in $\underline{\text{Table 15 of revised paper}}$, where we train one vortex model on one single video sequence and generalize it to others. Here are part of the results. We can find that Vortex degenerates a lot.
>
> | 64 × 64 Navier-Stokes Dataset | Relative L2 |
> | ----------------------------- | ----------- |
> | Vortex                        | 8.1379      |
> | HelmSim                       | 0.1261      |
>
> #### **Section 2: Interpretability Claim**
>
> > **Q4:** Does the Helmholtz decomposition significantly improve interpretability compared to baseline models?
>
> As we keep emphasizing in defining "Helmholtz dynamics" in $\underline{\text{Section 3.1}}$, HelmSim not only learns the velocity field but also estimates the potential and stream functions of fluid. We believe that this design surpasses other baselines in learning "physically interpretable evidence", not just the velocity fields.
>
> > **Q5:** In Figure 4, the model predicts non-zero velocity outside the boundary.
>
> Sorry for this misleading visualization. Since the Spreading Ink dataset is only within the circle boundary, we only compute the loss within the boundary. Thus, the part outside of the boundary is without restriction.
>
> We have added the implementation details in the $\underline{\text{Appendix A of revised paper}}$.
>
> #### **Section 3: Multiscale modeling**
>
> > **Q6:** The aggregation operation after "Integration" needs further clarification.
>
> We conduct this operation following U-Net. As per the reviewer's suggestion, we have included details in the $\underline{\text{Appendix A.2 of revised paper}}$.

---

> > ### Author Response · Authors · 2023-11-18
> > **Response to Reviewer Mqos [Part 2]**
> >
> > #### **Section 4: Comparing Baselines**
> >
> > > **Q7:** Some missing experimental results with cited baselines.
> >
> > As we stated in $\underline{\text{Baseline paragraph of Section 4}}$, Vortex, DARTS, PWC-Net with fluid Refinement and MWT are not suitable for some of the benchmarks, which means their performance will degenerate seriously or the running time is extremely slow if we stiffly use them to all benchmarks. But, as per the reviewer's request, we still provide the following experiments to ensure transparency.
> >
> > As presented in the following table, you can find that these methods fail in part of benchmarks. Given their official papers are not designed for these tasks, we only included these results in the Appendix to avoid potential misguidances to readers in judging these models when they see the bad results or poor efficiency.
> >
> > More detailed analysis for each baseline and their corresponding showcases are also included in $\underline{\text{Appendix D of revised paper}}$. Here are some key points:
> >
> > - Vortex: we have to re-train different models for every video sequence, which is unacceptable. Here we provide the results generalized from one model trained on one single video.
> > - DARTS: The prediction results are blurry. Also, it will take over 10 seconds for each inference, while other deep models only take less than 0.1 second.
> > - PWC-Net: It only estimate the velocity. To enable this model forecasting capability, we add a U-Net to utilize it estimated velocity field.
> > - MWT: Since it models the video sequence based on periodicity, the generated results contain weird texture in Spreading Ink.
> >
> > | Relative L2 | 64 × 64 Navier-Stokes | Bounded N-S | Sea Temperature | Spreading Ink (Video 3) |
> > | ----------- | --------------------- | ----------- | --------------- | ----------------------- |
> > | Vortex      | 8.1379                | 1.6259      | 4.9302 / 0.1796 | 5.973 / 0.1718 / 0.0150 |
> > | DARTS       | 0.8046                | 0.1820      | 0.3308 / 0.1094 | 4.940 / 0.1601 / 0.0130 |
> > | PWC-Net     | 0.1765                | 0.0729      | 0.1805 / 0.0406 | 5.341 / 0.1591 / 0.0128 |
> > | MWT         | 0.1586                | 0.1407      | 0.2075 / 0.051  | 1.521 / 0.1775 / 0.0160 |
> > | HelmSim     | 0.1261                | 0.0652      | 0.1704 / 0.0368 | 5.208 / 0.1584 / 0.0127 |
> >
> > > **Q8:** Compare the Helmholtz decomposition with a Clifford Layer (Brandstetter et al., 2022)
> >
> > As per the reviewer's request, we have added Clifford Layer as a baseline in all datasets. We report Relative L2 for Navier-Stokes dataset and Bounded N-S dataset, MSE and relative L2 for Sea Temperature dataset, and Perceptual loss, Relative L2 and MSE for Spreading Ink.
> >
> > | Relative L2    | 64 × 64 / 128 × 128 / 256 × 256 Navier-Stokes | Bounded N-S | Sea Temperature | Spreading Ink           |
> > | -------------- | --------------------------------------------- | ----------- | --------------- | ----------------------- |
> > | Clifford Layer | 0.1841 / 0.1088 / 0.1759                      | 0.1071      | 2.2390 / 0.0606 | 2.469 / 0.1939 / 0.0124 |
> > | HelmSim        | 0.1261 / 0.0807 / 0.1310                      | 0.0652      | 0.1704 / 0.0368 | 1.464 / 0.1399 / 0.0065 |
> >
> > #### **Section 5: Minor Issues**
> >
> > > **Q9:** Ensure proper citation format for baseline models.
> >
> > Thanks for the suggestion. We have corrected all the citations of baselines in the $\underline{\text{revised paper}}$.
> >
> > > **Q10:** Make sure that symbols are well-defined with clear reference to their definitions.
> >
> > We have clarified the symbols of Equation (4) in the $\underline{\text{revised paper}}$.
> >
> > > **Q11:** Are there additional insights or reasons for employing multi-head integration beyond the expected capacity and performance improvements?
> >
> > Firstly, as we stated in **Q1**, the Helmholtz Dynamics is one of the primary sources of performance improvement.
> >
> > Secondly, when we try to implement physics insights into deep models, adopting well-acknowledged tools to ensure optimization is a natural way. Thus, we use the multi-head style design, which is widely used in Transformers to augment model capacity.
> >
> > If the reviewer does expect more insights, we think the multi-head design can make the model learn multiple dynamics of the fluid, which can decompose the complex dynamics into multiple complementary components. But this is still under verification.

---

> > > ### Author Response · Authors · 2023-11-20
> > > **Request of Reviewer's attention and feedback**
> > >
> > > Dear Reviewer,
> > >
> > > This is a kind reminder that the 12 days reviewer-author discussion period only left 3 days. Please let us know if our response has addressed your concerns.
> > >
> > > Following your suggestion, we have answered your concerns and improved the paper in the following aspects:
> > >
> > > - We have **added new ablations on with or without Helmholtz Dynamics and with or without multi-head** to prove improvement brought by learning Helmholtz Dynamics is more significant than multiple heads. We also rephrased the ablation part for better understanding.
> > > - We have **added Clifford Layer as a new baseline and reported missing baselines on all the datasets**.
> > > - We highlighted the **interpretability of our model as learning physics quantities beyond solely learning the velocity field** and clarified the misleading visualization.
> > >
> > > **In total, we have added more than 60 new experiments. All of these results have been included in the $\underline{\text{revised paper highlighted in blue}}$.**
> > >
> > > Thanks again for your valuable review. We are looking forward to your reply and are happy to answer any future questions.

---

> > > > ### Comment · Reviewer_Mqos · 2023-11-22
> > > > **Response to the authors**
> > > >
> > > > Thank you for your response and the comprehensive experiments. I appreciate the authors' efforts in addressing my comments and questions; most of my doubts have been resolved. Given these efforts, I have decided to update my rating to 6.

---

> > > > > ### Author Response · Authors · 2023-11-22
> > > > > **Many thanks for your response and raising score**
> > > > >
> > > > > Sincerely thanks for your dedication and valuable suggestions, which have inspired us to improve our paper substantially.
> > > > >
> > > > > We also appreciate that you acknowledge our effort in rebuttal. We cannot accomplish this without your detailed review.

---

### Official Review · Reviewer_eKS7 · 2023-11-02

**Soundness:** 3 good
**Presentation:** 2 fair
**Contribution:** 2 fair
**Rating:** 6
**Confidence:** 5

**Summary:**

Inspired by the Helmholtz theorem, the authors designed the HelmDynamic block to learn the Helmholtz dynamics instead of the velocity field. The proposed HelmSim model can integrate learned dynamics along temporal dimensions in multiple spatial scales to yield future fluid predictions. The proposed method achieves SOTA performance in simulation and real-world datasets.

**Strengths:**

Significance The author proposes the HelmSim with the HelmDynamic block to capture Helmholtz dynamics. By integrating learned dynamics along temporal dimensions through Multiscale Integration Network. HelmSim can predict the future fluid with physically interpretable evidence.

Novelty: Inspired by the Helmholtz decomposition theorem, the author proposed a block to predict the potential and stream function independently instead of directly learning the velocity field.

Clarity and quality: The writing is in general clear to flow, and the reported result beats other baseline models

**Weaknesses:**

The ablation study is not comprehensive and convincing. There are some inaccurate statements—details in the questions part.

**Questions:**

1. There is no general formulation of stream function in 3D, how will you generalize the proposed method for 3D cases?

2. What are the Reynold numbers for the cases？ They look like laminar flows. How is your method's performance on turbulence?

3. Missing movies for the predictions to check for temporal coherence.

4. In the ablation study part, the directly learned velocity is not too bad. The error map looks similar except for a narrow region. What is the L2 error with and without the Helmholtz dynamics? What is the computational overhead for training with Helmholtz dynamics? Moreover, the error map looks the same for including and not including the boundary condition terms, making it hard to understand the benefits of including the boundary terms.  Can you also include the ablation study for the multihead, multiscale structure of your proposed model to demonstrate their effects?

5. The author mentioned that 3D fluid field is hard to observe, so they focus on the 2D cases, which is a false statement. There are many advanced techniques to observe the 3D fluid fields like PIV. It is more like the limitation of the current method that only works on 2D cases.

---

> ### Author Response · Authors · 2023-11-18
> **Response to Reviewer eKS7 [Part 1]**
>
> We would like to sincerely thank Reviewer eKS7 for providing a detailed review and insightful suggestions.
>
> > **Q1:** How will you generalize the proposed method for 3D cases?
>
> According to the official formalization of Helmholtz decomposition ${\mathbf{F}}(\mathbf{r}) = \nabla\Phi(\mathbf{r}) + \nabla \times {\mathbf{A}}(\mathbf{r}), \mathbf{r} \in \mathbb{V}$, the stream function for a 3D fluid field is a 3D vector ${\mathbf{A}}(\mathbf{r}) = (\mathbf{A}_x(\mathbf{r}), \mathbf{A}_y(\mathbf{r}), \mathbf{A}_z(\mathbf{r}))$. In 2D cases, we set the the velocity component on the $z$-axis to zero, that is $\mathbf{A}_x(\mathbf{r})=\mathbf{A}_y(\mathbf{r})=0$. To extend HelmSim to 3D fluid, the key point is to change the HelmDynamic block into learning 3D potential and stream functions $\widehat{\Phi}\in\mathbb{R}^{1\times D \times H\times W}$ and $\widehat{\mathbf{A}}\in\mathbb{R}^{3\times D \times H\times W}$, where $D$ is the additional depth dimension of 3D fluid. Then, following the Helmholtz decomposition presented in $\underline{\text{Eq. (1) of revised paper}}$, we can easily obtain the inferred 3D vector velocity field, thereby enabling HelmSim to achieve the velocity-aware 3D fluid prediction.
>
> > **Q2:** What are the Reynold numbers for the cases? How is your method's performance on turbulence?
>
> Sorry for this missing information. The Reynold number for the Navier-Stokes dataset is about $10^4$, and for the Bounded N-S dataset is about 300.
>
> To better evaluate the performance of the turbulence dataset, we experimented with a turbulence dataset provided by TF-Net [1]. We strictly followed its setting and reported the RMSE for predicting future 20 timesteps. In this dataset, Helmsim still perfroms best. These results have also been included in $\underline{\text{Appendix D of revised paper}}$.
>
> | Turbulence Dataset | HelmSim     | FNO     | UNet    | MWT     | U-NO    | LSM     | TF-Net  |
> | ------------------ | ----------- | ------- | ------- | ------- | ------- | ------- | ------- |
> | RMSE               | **1042.38** | 1187.44 | 1062.13 | 1276.64 | 3276.09 | 1069.26 | 1061.78 |
>
> Reference:
>
> [1] Wang, Rui, et al. "Towards physics-informed deep learning for turbulent flow prediction." KDD. 2020.
>
> > **Q3:** Missing movies for the predictions to check for temporal coherence.
>
> Following the reviewer's suggestion, we have provided the prediction movies in the $\underline{\text{revised supplementary material}}$, where HelmSim achieves favorable temporal coherence.

---

> > ### Author Response · Authors · 2023-11-18
> > **Response to Reviewer eKS7 [Part 2]**
> >
> > > **Q4:** Add more ablation experiments.
> >
> > **We have provided a comprehensive ablation study with quantitive results in $\underline{\text{Appendix B of original submission}}$. Most of the ablations mentioned by the reviewer have been included in that section.**
> >
> > (1) "The directly learned velocity is not too bad. What is the L2 error with and without the Helmholtz dynamics? What is the computational overhead for training with Helmholtz dynamics?"
> >
> > We have provided the relative L2 in $\underline{\text{Table 9 of original submission}}$. As per the reviewer's request, we newly added the computational overhead comparison, where learning Helmholtz Dynamics can significantly boost the model performance with marginal extra computation costs.
> >
> > | 64 × 64 Navier-Stokes Dataset      | Relative L2 | GPU memory (MiB) | Running Time (s/epoch) |
> > | ---------------------------------- | ----------- | ---------------- | ---------------------- |
> > | Directly learning velocity         | 0.1412      | 14.86            | 72.18                  |
> > | Learning Helmholtz Dynamics (Ours) | **0.1261**  | 16.30            | 80.20                  |
> >
> > (2) "The error map looks the same for including and not including the boundary condition terms, making it hard to understand the benefits of including the boundary terms."
> >
> > We also have provided the relative L2 for this ablation in $\underline{\text{Table 10 of original submission}}$. To highlight the benefits of incorporating boundary conditions, we also zoom in the essential area in $\underline{\text{Figure 10 of revised paper}}$.
> >
> > | Bounded N-S Dataset                 | Relative L2 | GPU memory (MiB) | Training Time (s/epoch) |
> > | ----------------------------------- | ----------- | ---------------- | ----------------------- |
> > | Not including boundary condition    | 0.0846      | 26.98            | 226.20                  |
> > | Including boundary condition (Ours) | **0.0652**  | 29.48            | 267.63                  |
> >
> > (3) "Include the ablation study for the multihead, multiscale structure of your proposed model."
> >
> > Please see $\underline{\text{Table 8 of original submission}}$ for these ablations. To trade-off efficiency and perfromance, we choose 4 heads and 3 scales as the offical configuration to 64$\times$64 Navier-Stokes Dataset.
> >
> > | 64 × 64 Navier-Stokes Dataset | 1 head | 4 head     | 8 head | 16 head |
> > | ----------------------------- | ------ | ---------- | ------ | ------- |
> > | Relative L2                   | 0.1344 | **0.1261** | 0.1279 | 0.1249  |
> > | Training time (s/epoch)       | 59.69  | 81.20      | 120.86 | 171.97  |
> >
> > | 64 × 64 Navier-Stokes Dataset | 2 scale | 3 scale    | 4 scale | 5 scale |
> > | ----------------------------- | ------- | ---------- | ------- | ------- |
> > | Relative L2                   | 0.1514  | **0.1261** | 0.1361  | 0.1330  |
> > | Training time (s/epoch)       | 64.43   | 81.20      | 99.83   | 120.06  |
> >
> > > **Q5:** "It is more like the limitation of the current method that only works on 2D cases."
> >
> > Thanks for the reviewer's suggestion. We have rephrased the statement about 3D fluid.
> >
> > But we have to point out that most of the recent neural fluid solvers only focus on the 2D fluid simulation, such as our baselines FNO, LSM, MWT, U-NO, and Vortex. 3D fluid simulation is a quite challenging problem. Following these well-acknowledged models, we mainly experiment with the 2D fluid.
> >
> > As we discussed in **Q1**, HelmSim can be naturally generalized to 3D fluid by learning 3D potential and stream functions. Given the mainstreaming experiments of our scope, we would like to leave this as the future work.

---

> ### Author Response · Authors · 2023-11-22
> **We are anticipating your feedback**
>
> Dear Reviewer,
>
> Many thanks for your valuable and detailed review.
>
> Following your suggestion, we have answered your concerns and improved the paper in the following aspects:
>
> - We have **added 1 new dataset (Turbulence) for all 7 models including TF-Net (KDD 2020)** to demonstrate our performance on turbulence.
> - We **recalled and rephrased the ablations provided in the original submission** to demonstrate the effectiveness of our design for boundary condition, multiscale and multihead.
> - We **clarified the generalization direction of HelmSim to 3D fluid** and highlighted that **most of the recent neural fluid solvers only focus on the 2D fluid simulation**, such as our baselines FNO, LSM, MWT, U-NO, and Vortex. 3D fluid simulation is a quite challenging problem, and we leave it to future work.
> - We **added movies of predictions** in supplementary materials for intuitive comparison.
>
> In total, we have added more than 60 new experiments. **All of these results have been included in the $\underline{\text{revised paper}}$.**
>
> We kindly remind you that **the reviewer-author discussion phase will end in 24 hours. After that, we may not have a chance to respond to your comments.**
>
> Sincere thanks for your dedication! We are looking forward to your reply.

---

> > ### Comment · Reviewer_eKS7 · 2023-11-23
> > **Thanks for the extensive rebuttal**
> >
> > I have increased my score to 6

---

> > > ### Author Response · Authors · 2023-11-23
> > > **Thanks for your response and raising score**
> > >
> > > Dear reviewer,
> > >
> > > We appreciate that you acknowledge our effort in rebuttal. Sincerely thanks for your valuable review and response, which helped us a lot in improving the presentation and experiments in our paper.
> > >
> > > Authors

---

### Author Response · Authors · 2023-11-18
**Summary of Revisions**

We sincerely thank all the reviewers for their insightful reviews and valuable comments, which are instructive for us to improve our paper further.

This paper presents HelmSim, a physics-interpretable fluid prediction model. **Instead of directly learning velocity fields, we prpose to learn the Helmholtz dynamics, which decomposes the intricate dynamics by learning inherent physics quantities: potential and stream functions drived from the Helmholtz theorem.** Experimentally, HelmSim surpasses 7 baselines including traditional numerical method and advanced deep learning methods on 4 typical datasets, covering **both synthetic and real-world datasets with known or unknown boundary conditions.**

The reviewers held some positive opinions of our paper, especially in our problem setup and experiments, in which the proposed method **“addresses an important problem concerning partially-observed dynamical systems“**. Also, we conduct **“extensive testing of their methods across a wide range of datasets”** and **“show the capability of the models, especially with the inclusion of real-world datasets"**. Moreover, **"the reported result beats other baseline models".**

The reviewers also raised insightful concerns. We made every effort to address all the concerns by providing detailed clarification and requested results. Here is the summary of the major revisions:

- **Ablation study (Reviewer Mqos, eKS7, 2nz9, rfHe):** We ran 9 types of ablations and rewrote the ablation part, Appendix B,  to comprehensively verify the effectiveness of each design in our model. The experiment results present that our design in HelmDynamic block is essential and bring the primary improvement.
- **Compare to FNO under the same parameter size (Reviewer 2nz9 and rfHe):** We claify that all the baselines are implemeted by their offical configuration. To ensure a fair comparison, we also enlarge the model size of FNO. We can find that even though FNO is with sufficient parameter, it still performs worse than HelmSim.
- **How to generalize HelmSim to 3D fluid (Reviewer eKS7 and 2nz9)**: We demonstrate that HelmSim can be naturally generalized to 3D fluid by learning 3D potential and stream functions. A new section, Appendix F, is added to disscuss this topic.
- **Question about experiment setting (Reviewer eKS7 and rfHe):** We report the Reynold numbers of benchmarks and add a new experiment on turbulent dataset. Besides, we also highlight that our experiment strictly follow the previous work.
- **Provide movies for comparison (Reviewer eKS7):** We have included movies for all baselines in the new supplementary materials, where HelmSim acheives favorable temporal coherence.
- **Complete the comparison of all baseline in all benchmarks (Reviewer Mqos):** We highlight that these baselines are not suitable for part of the benchmarks. As per the reviewer's request, we still complete the experiments, where the baselines' performance drops seriously.
- **Add comparison with Clifford Layer (Reviewer Mqos):** We compare HelmSim with Clifford Layer in all benchmarks. HelmSim still surpasses this new baseline.
- **Compare with CG-based models (Reviewer rfHe):** We point out that our experiment settings are distinct from CG-based models, where the prediction is only based on the observed physics quantities and the velocity field is unaccessible. We also add a new paragraph in related work to discuss the reviewer mentioned works.

**After seven full days of experiments and work (with 8 A100 GPUs), we have added more than 60 new experiment results and 8 pages of new analysis to address the mentioned issues. All the revisions have been included in the $\underline{\text{revised paper highlighted in blue}}$.**

The valuable suggestions from reviewers are very helpful for us to revise the paper to a better shape. We'd be very happy to answer any further questions.

Looking forward to the reviewer's feedback.

---

### Comment · Area_Chair_Dxs7 · 2023-11-21
**Reviewers: Please respond to authors or update review**

Dear Reviewers,

The discussion phase will end tomorrow.  Could you kindly respond to the authors rebuttal letting them know if they have addressed your concerns  and update your review as appropriate? Thank you.

-AC

---

### Meta-Review · Area_Chair_Dxs7 · 2023-12-09

**Metareview:**

**Summary**  This work proposes a method, HelmSim, for simulating 2D fluids. Inspired by the Helmholtz theorem, the authors define a network block, the HelmDynamic block, which decomposes the flow into curl-free and divergence-free parts which are easier to solve by learning the potential and stream function respectively.  HelmSim uses the HelmDynamic block in a multiscale network to predict the flow.  The model is evaluated over 3 simulated flow datasets including turbulent flow and two real world dataset including sea surface temperature and spreading ink videos.

**Metareview** Reviewers found the method to be mildly novel. While this particular method is new, it is related to previous work exploiting Helmholtz decomposition which is not adequately cited.  The paper would benefit from a presentation which better positions it revative to prior work and is more clear about the task being solved. The experiments are a strong point of the paper which show substantial outperformance of several strong baselines over several diverse datasets. This comparison was strengthened during the revision which added new baselines, experimental details, fairer comparisons with parameter standardization, and better ablations.  However, despite the rebuttal there remain questions about the magnitude of the improvement, the choice of tasks (observations of advected quantities), and the length of the future rollout.  Additionally, variance of training runs is omitted and training for methods is over a fixed number of epochs.  Another concern involves how well the method would generalize to 3D and compare to other baselines for 3D fluid simulation.

**Justification For Why Not Higher Score:**

- some but limited novelty
- some but limited significance/magnitude of empirical results
- issues with experimental set up
- inadequate evaluation length
- presentation issues with respect to prior work and claimed contribution relative to demonstrated tasks

**Justification For Why Not Lower Score:**

N/A

---

### Decision · Program_Chairs · 2024-01-16

Reject